# An Orthogonal Learner for Individualized Outcomes in Markov Decision Processes

**Emil Javurek** *
LMU Munich & MCML
emil.javurek@lmu.de

**Valentyn Melnychuk**
LMU Munich & MCML
melnychuk@lmu.de

**Jonas Schweisthal**
LMU Munich & MCML
jonas.schweisthal@lmu.de

**Konstantin Hess**
LMU Munich & MCML
k.hess@lmu.de

**Dennis Frauen**
LMU Munich & MCML
frauen@lmu.de

**Stefan Feuerriegel**
LMU Munich & MCML
feuerriegel@lmu.de

## Abstract

Predicting individualized potential outcomes in sequential decision-making is central for optimizing therapeutic decisions in personalized medicine (e.g., which dosing sequence to give to a cancer patient). However, predicting potential outcomes over long horizons is notoriously difficult. Existing methods that break the curse of the horizon typically lack strong theoretical guarantees such as orthogonality and quasi-oracle efficiency. In this paper, we revisit the problem of *predicting individualized potential outcomes in sequential decision-making* (i.e., estimating Q-functions in Markov decision processes with observational data) through a causal inference lens. In particular, we develop a comprehensive theoretical foundation for meta-learners in this setting with a focus on beneficial *theoretical properties*. As a result, we yield a novel meta-learner called DR$Q$-learner and establish that it is: (1) doubly robust (i.e., valid inference under the misspecification of one of the models), (2) Neyman-orthogonal (i.e., insensitive to first-order estimation errors in the nuisance functions), and (3) achieves quasi-oracle efficiency (i.e., behaves asymptotically as if the ground-truth nuisance functions were known). Our DR$Q$-learner is applicable to settings with both discrete and continuous state spaces. Further, our DR$Q$-learner is flexible and can be used together with arbitrary machine learning models (e.g., neural networks). We validate our theoretical results through numerical experiments, thereby showing that our meta-learner outperforms state-of-the-art baselines.

## 1 Introduction

Predicting individualized potential outcomes in sequential decision-making is central for optimizing therapeutic decisions in personalized medicine (Feuerriegel et al., 2024). Typical examples are selecting dosage schedules for cancer patients (Zhao et al., 2009; Wang et al., 2012), scheduling just-in-time interventions in digital health (Liao et al., 2021; Battalio et al., 2021), or determining treatment schedules for chronic diseases (Shortreed et al., 2011; Matsouaka et al., 2014). In recent years, this problem has been increasingly studied using observational data (e.g., electronic health records) to avoid "exploration" and leverage the increasing availability of digital patient data (Allam et al., 2021; Bica et al., 2021).

Here, we focus on predicting individualized potential outcomes in Markov decision processes (MDPs), i.e., *estimating the Q-function from observational data*. This task has received much attention in off-policy reinforcement learning (e.g., Liu et al., 2018; Le et al., 2019; Uehara et al., 2020), where many approaches have focused on delivering new learners, with a focus on addressing the curse of horizon. However, comparatively little attention has been given to developing methods in a principled way with theoretical guarantees such as orthogonality or quasi-oracle efficiency.

---

*corresponding author

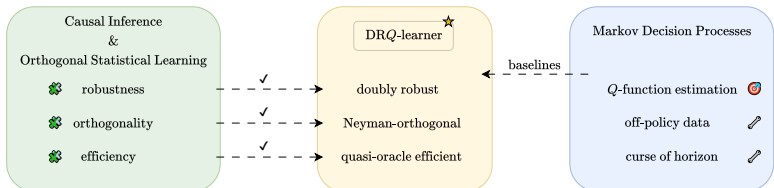

Figure 1: **Our work is located at the intersection of ① causal inference & orthogonal statistical learning and ② MDPs.** Our *problem setup* is in ②: we estimate Q-functions in MDPs from off-policy data. Baselines for this task break the curse of the horizon but typically lack strong theoretical guarantees. Our *method* adopts concepts from ①: we obtain a novel meta-learner called DR$Q$-learner that is doubly robust, Neyman-orthogonal, and quasi-oracle efficient.

In this paper, we study the problem of estimating $Q$-functions in MDPs from observation data through the theoretical lens of causal inference. In particular, we *develop a theoretical foundation* based on statistical orthogonality theory (Foster & Syrgkanis, 2019), which offers a novel perspective on this task (see Figure 1). For this, we first derive identifiability results and show that several of the existing baselines correspond to naïve plug-in learners, which are known to be biased. As a remedy, we next derive the efficient influence function of the training loss and use it to construct a debiased second-stage loss that is Neyman-orthogonal.

As a result, we obtain a novel meta-learner for this task, which we call **DR$Q$-learner**. The DR$Q$-learner enjoys several favorable theoretical properties: (1) it is *doubly robust*, which enables valid inference even under model misspecification; (2) it is *Neyman-orthogonal*, which makes it insensitive to first-order estimation errors in the nuisance functions; and (3) it achieves *quasi-oracle efficiency*, meaning it attains the same asymptotic performance as if the ground-truth nuisance functions were known. The DR$Q$-learner is applicable to settings with both discrete and continuous state spaces. Moreover, the DR$Q$-learner is flexible and can be used together with arbitrary machine learning models such as neural networks.

Our **contributions** are three-fold:[1]

- *New theoretical contributions.* We provide a theoretical framework of causal inference to $Q$-function estimation in MDPs. While causal inference has long been used to address statistical challenges in treatment effect estimation from observational data, we extend these ideas to formalize – and solve – the challenges of estimating $Q$-functions from observational data. In this setting, interventions induce a distributional shift between behavior and evaluation policies; although inverse propensity weighting (IPW) can address this, IPW suffers from exponentially decaying overlap in sequential settings (i.e., the *curse of horizon*), leading to instability from division by near-zero probabilities and making consistent estimation of potential outcomes impossible. By leveraging statistical orthogonality theory, we derive a novel meta-learner for *valid inference with favorable statistical properties*.

- *New method.* We propose the *first* meta-learner for $Q$-function estimation that is simultenously (i) *doubly robust*, (ii) *Neyman-orthogonal*, and quasi-oracle efficient. Hence, this is unlike methods that rely, for example, on IPW and are thus Neyman-orthogonal but fail to break the curse of horizon; the DR$Q$-learner avoids this issue and achieves all three properties while still addressing the curse of the horizon. Importantly, quasi-oracle efficiency of our method guarantees *convergence at the same rate as if oracle nuisance functions were known*. We thereby aim to make an important contribution to *reliable* inference in personalized medicine where strong theoretical guarantees are important.

- *Empirical performance.* The primary objective of our numerical experiments is to *validate our theoretical results*. Hence, we run various numerical experiments and show that the DR$Q$-learner is *especially effective for low overlap settings in line with our theory*. Overall, our results demonstrate state-of-the-art empirical performance.

---

[1]Code is available at `https://github.com/EmilJavurek/Orthogonal-Q-in-MDPs`.

## 2 RELATED WORK

We group our literature review along streams that are relevant: (1) We review theoretical foundations from causal inference and orthogonal statistical learning to motivate our method, and (2) discuss prior work on off-policy Q-function estimation in MDPs. The latter defines our problem setup, while the former shares parallels in terms of the overall methodological approach to formalize causal quantities. We provide an extended literature review in Appendix A.

**Causal inference and orthogonal learning:** Both the theory of orthogonal statistical learning (Foster & Syrgkanis, 2019) and semiparametric efficiency theory (van der Vaart, 1998) have been widely used to construct estimators with strong theoretical properties. Here, a particular focus is on influence-function-based estimators (Kennedy, 2022), with well-known examples such as targeted maximum likelihood estimation (TMLE) (Daniel Rubin, 2006), the DoubleML framework (Chernozhukov et al., 2018), and doubly robust approaches for off-policy policy value estimation (Kallus & Uehara, 2022; Shi et al., 2021). These techniques have been extended to the estimation of individualized treatment effects (Foster & Syrgkanis, 2019), leading to a broad class of orthogonal meta-learners (Kennedy, 2020; Nie & Wager, 2021; Morzywolek et al., 2023). Similarly, meta-learners have been proposed for estimating individualized treatment effect estimation over time (Frauen et al., 2025). However, works on individualized treatment effect estimation over time do ***not*** focus on the MDP setting and are well to known to suffer from the curse of horizon (Kallus & Uehara, 2022). Importantly, a similar theoretical framework for individualized potential outcome estimation in MDPs is still missing.

**Off-policy $Q$-function evaluation:** Several methods have been developed for estimating $Q$-function from MDPs in off-policy settings, that is, using observational data (e.g. Liu et al., 2018; Le et al., 2019; Uehara et al., 2020). A common theme in these works is to address the curse of horizon (e.g., Le et al., 2019; Uehara et al., 2020). We refer to Appendix A for a more detailed overview[2].

The above works have been developed typically outside of causal inference and thus without explicitly formalizing the underlying estimand as a causal quantity. One of our contributions is to link causal inference and $Q$-function evaluation from observational data by formalizing the underlying causal estimand. This allows us later to taxonomize prominent works from the literature based on the underlying adjustment strategy. For example, in our framework, existing works correspond to adjustment strategies based on inverse-propensity-weighting-like nuisances (e.g., Q-regression (Liu et al., 2018)) or implicit adjustment strategies based on (supervised learning) target construction (e.g., FQE (Le et al., 2019)). From our causal inference perspective, we later obtain new theoretical insights to understand the failing modes of existing methods. In particular, we show that several state-of-the-art methods suffer from so-called plug-in bias (Kennedy, 2022) and potential instability under model misspecification. To the best of our knowledge, more advanced adjustment strategies, which are commonly used in causal inference, are missing in the literature on $Q$-function evaluation. Consequently, no prior work has developed a Neyman-orthogonal meta-learner for off-policy $Q$-function estimation.

**Research gap:** To the best of our knowledge, a method for $Q$-function evaluation in MDPs with observational data that enjoys favorable theoretical properties - such as Neyman-orthogonality and quasi-oracle efficiency - is missing. As a remedy, we first reframe off-policy $Q$-function evaluation through the lens of causal inference and then develop a new meta-learner called DR$Q$-learner.

## 3 PROBLEM FORMULATION

**Notation:** We denote random variables by capital letters $S, A, R$ and their realizations by small letters $s, a, r$ from domains $\mathcal{S}, \mathcal{A}, \mathcal{R}$. Let $\mathbb{P}(S)$ denote a distribution of some random variable S, and let $p(S = s)$ be a corresponding density or probability mass function, and let $\mathcal{P}(\mathcal{S})$ denote the set of all probability distributions over $\mathcal{S}$. We write $\mathbb{E}_\pi[\cdot] := \mathbb{E}_{\mathbb{P}_\pi}[\cdot] = \int \cdot \, \mathrm{d}\mathbb{P}_\pi$ to denote expectation with respect to a distribution $\mathbb{P}_\pi$ arising from a stochastic process created by following the MDP with policy $\pi$ (equivalently, $\mathbb{E}_{x \sim P}[\cdot]$ when $x \sim P$).

---

[2]Many of these works focus on off-policy evaluation (and thus target *scalar average* outcomes), where methods for $Q$-function evaluation are often a necessary first step (e.g., Shi et al. (2021) propose a method for interval estimation that yields a Q-function evaluation as byproduct)

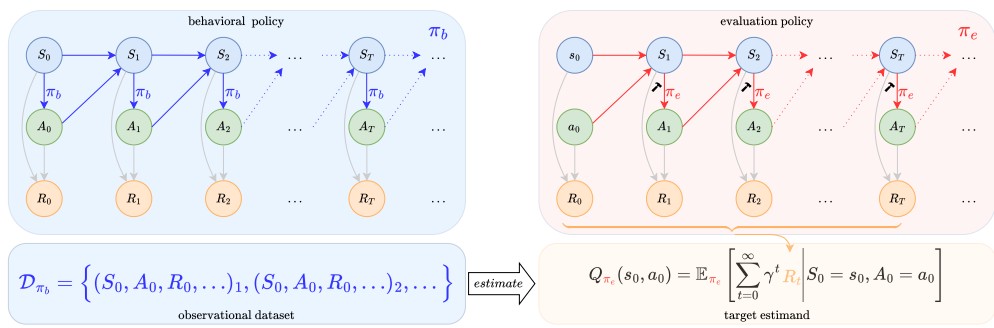

Figure 2: **Our task: we aim to estimate $Q_{\pi_e}$, a functional of the unobserved evaluation policy $\pi_e$ (right), from the observational dataset $\mathcal{D}_{\pi_b}$ from the behavioral policy $\pi_b$ (left).** A trajectory from a time-invariant Markov decision process (MDP) is determined by environment dynamics (gray) and by selecting actions according to a policy. We observe the MDP with $\pi_b$ (top left), while a potential MDP with $\pi_e$ (top right) is unobserved. Our target estimand $Q_{\pi_e}$ must thus be estimated from available observational data $\mathcal{D}_{\pi_b}$.

**Data-generating process:** We consider the following definition of a time-invariant MDP, as is common in the literature and in many practical applications[3] (Uehara et al., 2020; Shi et al., 2021; Kallus & Uehara, 2022). Formally, a time-invariant MDP is given by tuple $\langle \mathcal{S}, \mathcal{A}, \mathcal{R}, p_r, p_s, \gamma \rangle$ with: (i) $\mathcal{S}$ is the state space that can be discrete or continuous; (ii) $\mathcal{A}$ is the action space; (iii) $\mathcal{R}$ is the reward space, (iv) $p_r : \mathcal{S} \times \mathcal{A} \to \mathcal{P}(\mathcal{R})$ is the reward distribution, (v) $p_s : \mathcal{S} \times \mathcal{A} \to \mathcal{P}(\mathcal{S})$ is the stochastic state transition distribution, and (vi) $\gamma \in (0, 1)$ is the discount rate for future rewards. A trajectory $\{(S_t, A_t, R_t)\}_{t \geq 0}$ is generated by following a stationary stochastic policy $\pi$: at time step $t$, a decision-maker in state $S_t = s \in \mathcal{S}$ selects an action $A_t = a \in \mathcal{A}$ with probability $\pi(A_t = a \mid S_t = s)$, a reward $R_t = r$ is observed according to the law $R_t \sim p_r(s, a)$, and one transitions to a new state $S_{t+1} = s', S_{t+1} \sim p_s(s, a)$.

In our data-generating process, we assume (i) that the time-invariant MDP model has the Markov property $\mathbb{P}(S_{t+1} = s \mid \{S_j, A_j, R_j\}_{0 \leq j \leq t}) = p_s(s \mid S_t, A_t)$ and (ii) that the conditional mean independence property holds, i.e., $\mathbb{E}[R_t \mid \{S_j, A_j, R_j\}_{0 \leq j \leq t-1}, S_t, A_t] = p_r(S_t, A_t)$. Together, the assumptions (i) and (ii) guarantee the existence of an optimal stationary policy (Puterman, 1994) and permit us to decompose a dataset of i.i.d. trajectories into one-step transitions, namely

$$\mathcal{D}_\pi = \{(S_{i,t}, A_{i,t}, R_{i,t}, S_{i,t+1})\}_{0 \leq t \leq T, 1 \leq i \leq n} = \{(S_j, A_j, R_j, \tilde{S}_{j+1})\}_{j=1}^{N=nT} = \{O_j\}_{j=1}^{N=nT}, \quad (1)$$

where we use $O = (S, A, R, \tilde{S})$ to denote observations.

**Key quantities:** Given an *observational* dataset from a behavioral policy $\mathcal{D}_{\pi_b} \sim \pi_b$, we are then interested in estimating outcomes under a different evaluation policy $\pi_e$. The **target estimand** is the *state-action value function* $Q_{\pi_e}$ of $\pi_e$, which is defined as the $\gamma$-discounted expected cumulative reward across trajectories generated according to the policy $\pi_e$, i.e.,

$$Q_{\pi_e}(s, a) \triangleq \mathbb{E}_{\pi_e}\left[\sum_{t=0}^{\infty} \gamma^t R_t \,\middle|\, S_0 = s, A_0 = a\right]. \quad (2)$$

See Figure 2 for a visual illustration of the estimation task. We also define a *state value function* $v_{\pi_e}(s) \triangleq \mathbb{E}_{A \sim \pi(\cdot \mid s)}[Q_{\pi_e}(s, A)]$. We further introduce **nuisance functions**[4] of the cumulative and stationary density ratio via

$$\rho_{l:t} \triangleq \prod_{k=l}^{t} \frac{\pi_e(A_k = a_k \mid S_k = s_k)}{\pi_b(A_k = a_k \mid S_k = s_k)}, \; w_{e/b}(s' \mid s, a) \triangleq \frac{\sum_{t=1}^{\infty} p_e(S_t = s' \mid S_0 = s, A_0 = a)}{p_b(S = s')}, \quad (3)$$

---

[3]Importantly, we can always incorporate any historical information into state variable, simply by concatenating both and creating an "augmented" state variable where $\tilde{S} = (S, H)$ where $H$ is any additional historical information we wish to store beyond any natural concept of state. The Markov framework will natively apply to such a setup. As such, the MDP setting can also be seen as a generalization of contextual bandits and some dynamic treatment regime (DTR) setups. Specifically, contextual bandits are a one-step MDP with no state transitions (or equivalently, all state transitions are equally possible, i.e., irrelevant). DTR setups usually operate on short time horizons since without assuming Markovianity (=MDP), they invariably also suffer from the curse of horizon. We refer to Appendix A for extended related work regarding DTR and the relevant overview materials.

[4]We call nuisance functions all auxiliary functions that are not of primary interest but are required for estimation.

respectively. The subscripts $e, b$ in $p_e, p_b$ are used to denote densities arising from following an MDP with an evaluation and behavioral policy, respectively. We collect the nuisances in a tuple $\eta = (\rho, w_{e/b})$.

### 3.1 CAUSAL INTERPRETATION

**Objective:** Given an *observational* dataset from a behavioral policy $\mathcal{D}_{\pi_b} \sim \pi_b$, we are then interested in estimating outcomes under a different evaluation policy $\pi_e$. Since data following $\pi_e$ is not observed, our target is a causal quantity. To formalize this, we use the potential outcomes framework (Neyman et al., 1923; Rubin, 1974) and denote the potential reward by $R[a]$, i.e., the reward that *would have been observed had action a been selected*. Then, $R[\pi_e] \triangleq \sum_{a \in \mathcal{A}} R[a] \pi_e(a \mid S)$ is the potential reward that would have been observed under the policy $\pi_e$ (Uehara et al., 2022). Hence, we are interested in estimating the potential state-action value had policy $\pi_e$ been followed:

$$\xi_{\pi_e}(s, a) \triangleq \mathbb{E}\left[R_0 + \sum_{t=1}^{\infty} \gamma^t R_t[\pi_e(\cdot \mid S_t)] \,\middle|\, S_0 = s, A_0 = a\right]. \tag{4}$$

The causal estimand $\xi_{\pi_e}(s, a)$ characterizes the *expected individualized potential outcomes* in sequential decision-making (e.g., the patient-specific outcome from a dosage schedule of anti-cancer drugs for a specific patient trajectory). If identification assumptions hold (see Appendix B), the *causal estimand* $\xi_{\pi_e}$ is identified as a *statistical estimand* $Q_{\pi_e}$ and can thus be estimated from the observational data (i.e. can be expressed as functional of only the observable distribution from following $\pi_b$). Below, we state the identification results in two ways: in Lemma 1, we take observational data at the level of trajectories, whereas, in Lemma 2, we take the observational data at the level of one-step transitions. While the first approach is more straightforward, the second allows us to later break the curse of horizon when we develop the DR$Q$-learner.

**Lemma 1** (Identification *over trajectories*). *Under Assumptions (1)–(3) from above, the causal estimand in Eq. (4) is identifiable from the observed data of trajectories via*

$$\xi_{\pi_e}(s, a) = Q_{\pi_e}(s, a) = \mathbb{E}_{\pi_b}\left[R_0 + \sum_{t=1}^{\infty} \gamma^t \rho_{1:t} R_t \,\middle|\, S_0 = s, A_0 = a\right]. \tag{5}$$

*Proof.* See Appendix D.4. □

**Lemma 2** (Identification *over one-step transitions*). *Under Assumptions (1)–(3), the causal estimand in Eq. (4) is identifiable from the observed data of one-step transitions via* $\xi_{\pi_e}(s, a) = Q_{\pi_e}(s, a) = f(s, a)$, *where $f$ is the unique solution (unique up to equality almost everywhere) to the Bellman equation for $\pi_e$, i.e.,*

$$f(s, a) = \mathbb{E}\left[R + \gamma \mathbb{E}_{\tilde{A} \sim \pi_e(\cdot \mid \tilde{S})}[f(\tilde{S}, \tilde{A})] \,\middle|\, S = s, A = a\right]. \tag{6}$$

*Proof.* See Appendix D.4. □

While the derivations of the above identifiability results are straightforward, our aim behind these is to cast the target explicitly as a causal estimand. In the following section, we build on these identification Lemmas and recast existing $Q_{\pi_e}$ estimation algorithms as causal plug-in learners.

## 4 A ROADMAP TO ORTHOGONAL LEARNING

To derive our method for estimating $Q_{\pi_e}$ from observational data $\mathcal{D}_{\pi_b}$, we proceed in three steps: ① We first leverage the above identifiability results to construct simple plug-in learners (Section 4.1). We show that these plug-in learners recover existing methods from the literature, namely, $Q$-regression (Liu et al., 2018) and FQE (Le et al., 2019). However, *plug-in learners have inherent limitations such as so-called plug-in bias* (Kennedy, 2022). This serves two-fold: to formalize the drawbacks of existing methods theoretically (using the lens of the potential outcomes framework) and to motivate an alternative estimation strategy. ② We then sketch out the idea behind designing two-stage meta-learners based on Neyman-orthogonal losses (Section 4.2). ③ Finally, we then present our new Neyman-orthogonal meta-learner called DR$Q$-learner (Section 5). To do so, we leverage semiparametric efficiency theory and derive the efficient influence function. We also show that our new meta-learner has several favorable theoretical properties, namely, double robustness, Neyman-orthogonality, and quasi-oracle efficiency. We provide an overview of the different learners in Figure 3.

### 4.1 WHY PLUG-IN LEARNERS ARE SUB-OPTIMAL

The identification results from above (i.e., Lemma 1 and Lemma 2) give immediately rise to two naïve plug-in estimators. However, as we show later, each comes with inherent limitations.[5]

• **IPTW plug-in learner:** A straightforward way to obtain an estimator of $Q_{\pi_e}$ is to take the identification result based on Lemma 1 (i.e., right-hand side of Eq. (5)) and "plug-in" an estimated cumulative density ratio nuisance $\hat{\rho}_{1:t}$. This yields

$$\hat{Q}_{\pi_e}^{\text{IPTW}}(s,a) = \frac{1}{n}\sum_{i=1}^{n}\left[\left(R_{i,0} + \sum_{t=1}^{\infty}\gamma^t\hat{\rho}_{i,1:t}R_{i,t}\right)\mathbb{I}\{S_{i,0}=s, A_{i,0}=a\}\right], \qquad (7)$$

which involves the density ratio $\hat{\rho}_{1:t}$ and thus captures the inverse probability of treatment weighting (IPTW). When we then generalize the estimator from a tabular point-wise solution to learning the best model $\hat{g}$ from a restricted model class $\mathcal{G}$, we yield

$$\hat{Q}_{\pi_e} = \hat{g} = \arg\min_{g\in\mathcal{G}}\frac{1}{n}\sum_{i=1}^{n}\left[\sum_{t\geq 0}\gamma^t\hat{\rho}_{i,1:t}\left(Y_{i,t} - g(S_{i,t},A_{i,t})\right)^2\right] \text{ for } Y_{i,t} = \sum_{t'\geq t}\gamma^{t'-t}\hat{\rho}_{i,(t+1):t'}R_{i,t'}, \quad (8)$$

which corresponds exactly to $Q$-regression (Liu et al., 2018).[6] A specific limitation of the IPTW plug-in learner (=$Q$-regression) is that it suffers from the curse of horizon as a consequence of using the *cumulative* density ratio nuisance $\hat{\rho}_{1:t}$.

• **Recursive plug-in learner:** An alternative is to use the second identification result from Lemma 2. Analogous to the technique used in the identification proof (see Appendix D.4), we can recursively obtain an estimator $\hat{Q}_{k+1}$ by "plugging-in" into the right-hand side of Eq. (6) the previous estimator $\hat{Q}_k$. Formally, we have

$$\hat{Q}_{\pi_e}^{\mathfrak{R}} = \lim_{k\to\infty}\hat{Q}_k \quad\text{for}\quad \hat{Q}_{k+1}(s,a) = \frac{1}{N}\sum_{i=1}^{N}\left[\left(R_i + \gamma\mathbb{E}_{\tilde{A}\sim\pi_e(\cdot|\tilde{S}_i)}[\hat{Q}_k(\tilde{S}_i,\tilde{A})]\right)\mathbb{I}\{S_i=s, A_i=a\}\right]. \quad (9)$$

This yields an estimated solution to the empirical approximation of Eq. (6). Generalizing this approach to a minimization over a model class $\mathcal{G}$, we yield

$$\hat{Q}_{\pi_e} = \hat{g} = \lim_{k\to\infty}\hat{g}_k \quad\text{for}\quad \hat{g}_{k+1} = \arg\min_{g\in\mathcal{G}}\frac{1}{N}\sum_{i=1}^{N}\left[\left(R_i + \gamma\mathbb{E}_{\tilde{A}\sim\pi_e(\cdot|\tilde{S}_i)}[\hat{g}_k(\tilde{S}_i,\tilde{A})] - g(S_i,A_i)\right)^2\right], \quad (10)$$

which corresponds exactly to the FQE baseline (Le et al., 2019)). While the recursive plug-in learner (=FQE) breaks the curse of horizon, its recursive fitting procedure may lead to unpredictable failure modes or even divergence (see the problem of deadly triad in, e.g., Sutton & Barto (2018)).

⇒ *Fundamental problems of plug-in learners:* Both plug-in learners suffer from so-called *plug-in bias* (Kennedy, 2022): that is, *errors in the nuisance estimates directly propagate to the causal estimand*. In contrast, we now derive our Neyman-orthogonal meta-learner that *eliminates first-order bias from the nuisance functions*. Hence, bias from nuisance function estimates propagates into the final estimand only via higher-order errors.

Figure 3: **Comparison.** After observing the data $\mathcal{D}_{\pi_b}$, the learner-specific nuisance functions are estimated first, followed by the actual estimand. ☆= our DR$Q$-learner. Learners suffering from plug-in bias are marked with ✗.

### 4.2 INTUITION BEHIND TWO-STAGE META-LEARNERS

To resolve issues from plug-in bias, we later develop a two-staged meta-learner (see Fig. 3). The basic idea is: ① In the **first stage**, the nuisances

---

[5] For ease of exposition, we adopt the nomenclature for naming different methods based on causal inference literature, but later state the corresponding names of the benchmarks in the literature.

[6] To see why this is a generalization of the tabular $\hat{Q}_{\pi_e}^{\text{IPTW}}$, consider the case of having a free parameter $\theta$ for each possible point evaluation. The learned minimizer $\hat{g}$ is then nothing else than a point-wise solution to the estimating equation $\nabla_\theta\mathcal{L}(\theta)=0$, which will simply equate $\hat{g}(s,a) = \frac{1}{n}\sum_{i=1}^{n}Y_{i,t}\cdot\mathbb{I}\{S_{i,t}=s, A_{i,t}=a\}$.

are estimated, yielding some estimate $\hat{\eta}$. ② In the **second stage**, the target $g$ with true value $g^*$ is estimated by empirical risk minimization (ERM) over a risk $\mathcal{L}$ via

$$\hat{g} = \arg\min_{g \in \mathcal{G}} \mathcal{L}(\hat{\eta}, g). \tag{11}$$

Here, one seeks a learner (second-stage loss) with small error despite learning $\hat{g}$ with the estimated nuisance $\hat{\eta}$ carrying first-stage estimation error. However, deriving such a second-stage loss is non-trivial.

A common feature for the second-stage loss is to employ *Neyman-orthogonal loss functions* (Chernozhukov et al., 2018), which (in population) satisfy the property

$$D_\eta D_g \mathcal{L}(g^*, \eta)[\hat{g} - g, \hat{\eta} - \eta] = 0, \tag{12}$$

where $D_g$ and $D_\eta$ are directional (Gateaux) derivatives in function space (Foster & Syrgkanis, 2019). Informally, orthogonality means the gradient of the loss $D_g \mathcal{L}$ (i.e., the estimating function, or also known as the *score*) is insensitive to small perturbations in the nuisances around their oracle value $\eta$, such as those arising from nuisance estimation error.[7]

# 5 OUR DR$Q$-LEARNER

We proceed in three steps: ① We first derive our Neyman-orthogonal loss. ② Next, we show the Quasi-oracle efficiency and double-robustness properties of our loss. ③ Finally, we elaborate on the practical implementation.

## 5.1 THEORETICAL RESULTS

We denote our Neyman-orthogonal loss by $L^3_{\pi_e}(\eta, g)$, which we formally derive in Theorem 1. Therein, we derive and employ the *efficient influence function* (EIF). With the perspective of classical semiparametric inference, we replace the ERM *estimate* of the population risk with a debiased estimator based on the EIF. Under standard regularity conditions, the resulting population analogue is Neyman-orthogonal. Hence, by deriving the EIF of a standard MSE population risk, we obtain our main result, namely, the Neyman-orthogonal loss $L^3_{\pi_e}(\eta, g)$.

**Theorem 1** (*Neyman-orthogonality*)**.** *The loss*

$$L^3_{\pi_e}(\eta, g) = \mathbb{E}_{O' \sim p_b}\left[ \sum_a \pi_e(a \mid S')\left(\phi_1 - g(S', a)\right)^2 \right] + \mathbb{E}_{O' \sim p_b, s \sim p_b(s)}\left[ \sum_a \pi_e(a \mid s)\left(\phi_2 - g(s, a)\right)^2 \right] \tag{13}$$

*where*

$$\phi_1 = 2\frac{\mathbb{I}(A' = a)}{\pi_b(A' \mid S')}\left\{ R' + \gamma v_{\pi_e}(\tilde{S}') - Q_{\pi_e}(S', A') \right\} + Q_{\pi_e}(S', a), \tag{14}$$

$$\phi_2 = 2\frac{\pi_e(A' \mid S')}{\pi_b(A' \mid S')}w_{e/b}(S' \mid s, a)\left\{ R' + \gamma v_{\pi_e}(\tilde{S}') - Q_{\pi_e}(S', A') \right\} + Q_{\pi_e}(s, a) \tag{15}$$

*is Neyman-orthogonal w.r.t. all the nuisance functions $\eta = (\pi_b, w_{e/b}, Q_{\pi_e})$. For intuition on the form of the pseudo-outcomes $\phi_1, \phi_2$, we refer to Appendix D.1.1.*

*Proof.* We refer the reader to Appendix D.1 for formal proof. Here, to provide intuition, we show the efficient influence function of the standard MSE loss, $L^1_{\pi_e}(\eta, g)$, which is shown to be

$$\mathbb{IF}(L^1_{\pi_e}(\eta, g), O')$$

$$= \sum_a \pi_e(a|S')\left(Q_{\pi_e}(S', a) - g(S', a)\right)^2 - L^1_{\pi_e}(\eta, g) + 2\left\{ R' + \gamma v_{\pi_e}(\tilde{S}') - Q_{\pi_e}(S', A') \right\}\frac{\pi_e(A'|S')}{\pi_b(A'|S')} \tag{16}$$

$$\times \left[ Q_{\pi_e}(S', A') - g(S', A') + \mathbb{E}_{s, a \sim p_b(s)\pi_e(a|s)}\left[ (Q_{\pi_e}(s, a) - g(s, a))w_{e/b}(S'|s, a) \right]. \right]$$

Afterward, we derive the loss $L^3_{\pi_e}(\eta, g)$ with the debiasing procedure (and some algebraic manipulations). Neyman-orthogonality is then proved by taking the necessary derivatives. A formal and detailed derivation is in Appendix D.1. □

---

[7]For an extended discussion of orthogonal statistical learning, we refer to Appendix C.

Theorem 1 shows that our loss, $L^3_{\pi_e}(\eta, g)$, for estimating $Q_{\pi_e}$ is Neyman-orthogonal and, therefore, robust to nuisance estimation error. Finally, we prove our loss is quasi-oracle efficient and doubly robust.

**Theorem 2** (**Quasi-oracle efficiency**). *Under standard assumptions (see Foster & Syrgkanis (2019)), $L^3_{\pi_e}(\eta, g)$ achieves quasi-oracle efficiency, specifically, for $\hat{g} = \arg\min_{g \in \mathcal{G}} L^3_{\pi_e}(\hat{\eta}, g)$*

$$\|g^* - \hat{g}\|^2_{2, p_b \pi_e} \lesssim \|\Delta^2 \hat{\pi}_b\|^2_2 \|\Delta^2 \hat{Q}_{\pi_e}\|^2_2 + \|\Delta^2 \hat{w}_{e/b}\|^2_2 \|\Delta^2 \hat{Q}_{\pi_e}\|^2_2, \tag{17}$$

*where $x \lesssim y$ is taken to mean there exists a constant $C > 0$ such that $x \leq Cy$, the $\Delta k$ operator is defined as $\hat{k} - k^*$ for any function $k$, and $g^* = \arg\min_{g \in \mathcal{G}} L^3_{\pi_e}(\eta, g)$, which equals the true $Q_{\pi_e}$ provided the function class $\mathcal{G}$ is expressive enough to include it. Lastly, the norm weighting $p_b \pi_e$ in $\|\cdot\|^2_{2, p_b \pi_e}$ mirrors that of the loss.*

**Corollary 1** (**Double robustness**). *The learned approximation $\hat{g}$ is doubly robust. Specifically, if either $\Delta \hat{Q}_{\pi_e} \to 0$ or $\Delta \hat{\pi}_b \to \Delta \hat{w}_{e/b} \to 0$, then $\hat{g}$ is a consistent estimator of $g^*$, i.e., asymptotically $\|g^* - \hat{g}\|^2_{2, p_b \pi_e} = 0$.*

*Proof.* See Appendix D.3 for the proofs of both the theorem and the corollary. $\qquad\square$

The bound in Eq. (17) shows that the excess risk of $\hat{g}$ depends only on *products* of nuisance estimation errors. This means that even if one nuisance component (e.g. $\hat{Q}^1_{\pi_e}$) converges slowly, the overall estimator still converges at the fast rate of the better-estimated component. In other words, $\hat{g}$ behaves *as if oracles nuisances were used*, up to higher-order terms (cf. Foster & Syrgkanis, 2019; Nie & Wager, 2021). The estimation error is thus shielded from first-order nuisance misspecification and is only impacted through second-order interactions.

*Remark:* Our above theory is different from Shi et al. (2021) in the following ways: For the purpose of obtaining a tight confidence interval for OPE, Shi et al. (2021) have derived a point-wise iterative debiasing procedure for (in their view nuisance) $Q_{\pi_e}$ that, *when restricted to the discrete state setting with no model class $\mathcal{G}$ restrictions*, corresponds to our learner. We provide a more general solution that (i) is applicable to both continuous[8] and discrete state spaces, (ii) able to fit an estimator $\hat{g} \in \mathcal{G}$, and (iii) provides the theory necessary to show Neyman-orthogonality and quasi-oracle

---

**Algorithm 1** Our DR$Q$-learner for $Q_{\pi_e}$

**Input:** Observed dataset $\mathcal{D}_{\pi_b}$, class $\mathcal{G}$

**Output:** Doubly Robust estimator $\hat{Q}^{\text{DR}}_{\pi_e}$

1: // First stage (nuisance estimation)
2: $\hat{\pi}_b(a, s) \leftarrow \hat{\mathbb{P}}_b(A = a \mid S = s)$
3: $\hat{w}_{e/b}(s', s, a) \leftarrow \frac{\sum_{t=1}^{\infty} \hat{\mathbb{P}}_e(S_t = s' \mid S_0 = s, A_0 = a)}{\hat{\mathbb{P}}_b(s')}$
4: $\hat{Q}^1_{\pi_e} \leftarrow \hat{\mathbb{E}}_\pi \left[ \sum_{t=0}^{\infty} \gamma^t R_t \big| S_0 = s, A_0 = a \right]$
5: // Second stage (DR adjustment)
6: $\hat{Q}^{\text{DR}}_{\pi_e} = \arg\min_{g \in \mathcal{G}} \hat{L}^3_{\pi_e}((\hat{\pi}_b, \hat{w}_{e/b}, \hat{Q}^1_{\pi_e}), g)$
7: **Return:** $\hat{Q}^{\text{DR}}_{\pi_e}$

---

efficiency. Additionally, our derivation of the efficient influence function means that, for the discrete setting, we show that the estimator is efficient[9].

### 5.2 IMPLEMENTATION

**Pseudocode:** The pseudocode for our DR$Q$-learner is in Algorithm 1. (1) The first stage simply estimates the nuisance functions, namely, $\hat{\eta} = (\hat{\pi}_b, \hat{w}_{e/b}, \hat{Q}^1_{\pi_e})$. Notably, the nuisances include the target itself $Q_{\pi_e}$. (2) The aim of the second stage estimation is to refine the first stage estimate $\hat{Q}^1_{\pi_e}$ with a loss designed to bring favorable theoretical properties to the second stage refinement. Put differently, *we construct a meta-learner that in the first stage accepts any off-policy $Q_{\pi_e}$ estimation method and subsequently refines it*. Furthermore, we may choose to restrict the space of solutions $\mathcal{G}$ of the second stage and obtain the best projection of true $g^* \notin \mathcal{G}$ onto $\mathcal{G}$, for example, if we wish to obtain an interpretable solution.

**Implementation:** Our DR$Q$-learner is generally flexible and can be implemented with *arbitrary machine learning models* for estimating the nuisance functions as well as the second-stage. We provide details about the architectures and fitting process we use in our experiments in Appendix E.

---

[8]Note that the approach of Shi et al. (2021) cannot readily be extended to continuous settings since their point-wise debiasing step includes a Dirac delta function on the state. In a continuous setting, this is either zero or infinite, and thus not directly applicable.

[9]Meaning it achieves the semiparametric efficiency bound on asymptotic variance dictated by the EIF.

## 6 EXPERIMENTS

The primary goal of our experiments is not traditional benchmarking but rather to validate our theoretical results: ① that *our DRQ-learner outperforms the plug-in learners*; ② that our DRQ-learner is especially *effective in settings that benefit from Neyman-orthogonality* such as settings with low overlap; and ③ that *our theory is applicable to different function classes* including restricted model classes $\mathcal{G}$.

**Settings:** We consider the Taxi and Frozen Lake environments from the OpenAI Gym package (Brockman et al., 2016). We set our data-generating policy $\pi_b$ and our target evaluation policy $\pi_e$ as epsilon-greedy policies, $\pi_i \leftarrow \varepsilon\text{-greedy}(Q^*, \varepsilon_i)$ for $i \in \{e, b\}$ and for the optimal $Q^*$, which we acquire in an online fashion. We generate a dataset $\mathcal{D}_{\pi_b}$ of $n$ trajectories following $\pi_b$. We consider two settings: **(A)** when the model class $\mathcal{G}$ is left unrestricted, and **(B)** when the model class $\mathcal{G}$ is restricted to a simple linear model. For each setting, we conduct three sets of experiments: (1) We consider a varying dataset size $n \in [2000, \ldots, 6000]$. (2) By varying the discount factor $\gamma$, we alter the length of the horizon considered. Here, we vary the effective horizon[10] $h \triangleq \frac{1}{1-\gamma}$ in the range $h \in [3, \ldots, 20]$, or in other words, $\gamma \in [0.66, \ldots, 0.95]$. (3) By varying the greediness $\varepsilon_e \in [0.1, \ldots, 0.9]$ of the target evaluation policy, while holding $\varepsilon_b$ fixed, we can directly vary the degree of overlap between the dataset and the off-policy potential distribution whose $Q$-function we seek to estimate. We use a simple metric $\text{Overlap} = \sum_a \min(\pi_b(a), \pi_e(a))$ to quantify the level of overlap.

**Metric:** We evaluate the performance of all methods using $\text{rMSE}(\hat{Q}, Q_{\pi_e}) = \frac{\|\hat{Q} - Q_{\pi_e}\|_2^2}{\|Q_{\pi_e}\|_2^2}$. We report the mean ($\pm$ 1 standard error) over 5 runs with different seeds.

**Baselines:** As baselines, we implement standard $Q_{\pi_e}$ estimation methods of $Q$-regression (Liu et al., 2018) and FQE (Le et al., 2019). We have previously shown that these correspond to plug-in methods and should thus be inferior. Additionally, we implement Minimax $Q$-learning (MQL) (Uehara et al., 2020). For implementation details, see Appendix E

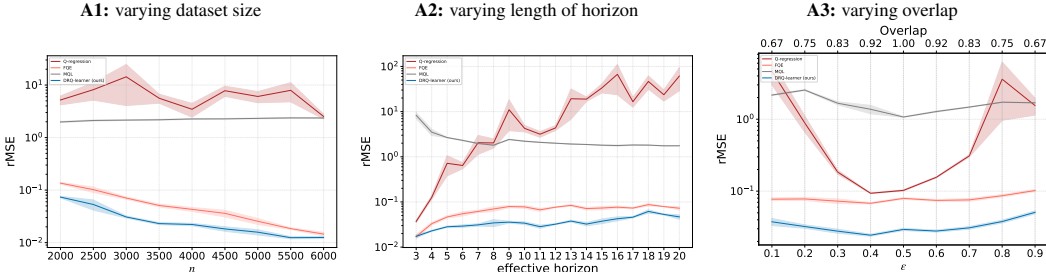

Figure 4: **Setting A – Taxi environment:** Unrestricted model class $\mathcal{G}$. The results *confirm the theoretical properties*: our DRQ-learner in blue is better than the plug-in learners in red/orange, robust for varying lengths of the horizon, and is especially effective for settings with low overlap.

**Results:** Results for Setting **A** (unrestricted) are in Fig. 4. Our DRQ-learner performs best across a variety of configurations. In particular, we confirm: ① *our method consistently outperforms the plug-in learners*. Further, our experiments show our method successfully incorporates the density ratio nuisance without degrading performance in the low overlap scenario (see Fig. A3). Hence, *the empirical results confirm our theoretical properties*. In particular, we confirm ② that our DRQ-learner is especially effective for long horizons and for low overlap settings in line with our theory. Results for Setting **B** (restricted) are in Fig. 5. Our method is highly effective and performs best for many settings, especially with low overlap[11]. Thereby, we confirm ③ that our theory is also applicable to restricted model classes.

---

[10]*Intuition:* Since $\sum_{t=0}^{\infty} \gamma^t = \frac{1}{1-\gamma}$, state-action values will have a magnitude of $\frac{1}{1-\gamma}$ times that of rewards. Instead of thinking of discounted rewards across an unbounded trajectory, we consider effectively taking a horizon of $h$ steps with undiscounted rewards.

[11]*Note about Figure 5:* At large $\varepsilon$, the target policy becomes nearly random, making Q-regression behave unusually well. This is because the density ratios shrink rather than explode, removing its usual instability. In this easy-nuisance regime the robustness and asymptotic benefits of DRQ-learner matter less, so its finite-sample performance does not strictly dominate.

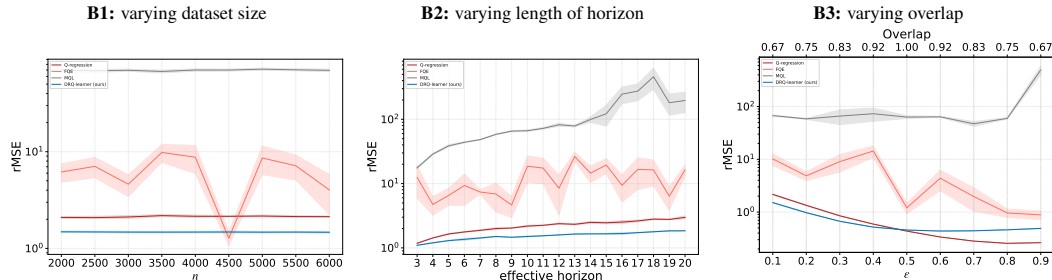

Figure 5: **Setting B – Taxi environment:** linear model class $\mathcal{G}$. The results confirm that *our theory and thus our DRQ-learner (in blue) are applicable to different (restricted) function classes*.

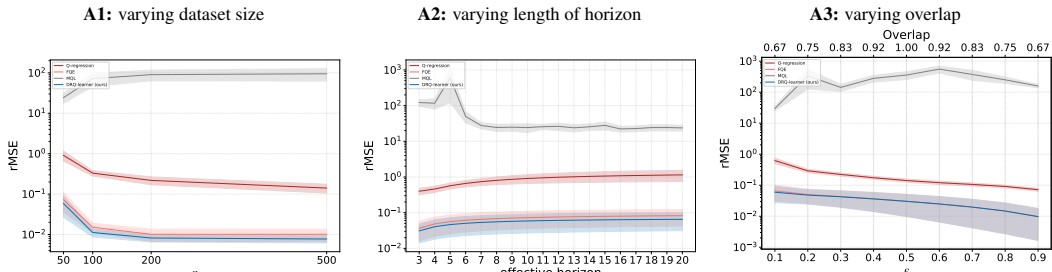

Figure 6: **Setting A – Frozen Lake environment:** Unrestricted model class $\mathcal{G}$. The results *confirm the theoretical properties*: our DR$Q$-learner in blue is better than the plug-in learners in red/orange, robust for varying lengths of the horizon.

**Conclusion:** In sum, our DR$Q$-learner is the first approach to jointly achieve double robustness, Neyman-orthogonality, and quasi-oracle efficiency. Thereby, we provide a principled and flexible foundation for *reliable* individualized decision-making in sequential settings. A particular advantage of our DR$Q$-learner is its flexibility to accommodate real-world constraints such as interpretability or fairness into the solution space $\mathcal{G}$.

## ACKNOWLEDGMENTS

This paper is supported by the DAAD programme Konrad Zuse Schools of Excellence in Artificial Intelligence, sponsored by the Federal Ministry of Research, Technology and Space. Additionally, this work has been supported by the German Federal Ministry of Education and Research (Grant: 01IS24082).

## ETHICS STATEMENT

Our work develops a theoretically principled approach, the DR$Q$-learner, for estimating individualized potential outcomes in MDPs from observational data. The primary goal of this work is to improve the *reliability* of decision-making algorithms, particularly in high-stakes settings such as personalized medicine. To promote reliable decision-making, we focus on identifiability results that transparently state the boundary conditions of our method and thus ensure when our method can be safely used.

*Potential benefits and risks:* Our method aims to enhance safe and effective individualized decision-making by providing more statistically reliable $Q$-function estimates even under model misspecification and low-overlap conditions. This can ultimately contribute to better treatment policies in healthcare and other critical applications. However, like any method that can be applied to personalized decision-making, misuse in inappropriate or sensitive settings could have unintended negative consequences (e.g., reinforcing biases present in observational data). We therefore emphasize that a successful application of our method requires domain expertise to ensure causal assumptions such as unconfoundedness and positivity, which are necessary for identifiability. Our work explicitly frames $Q$-function estimation as a causal inference problem, aligning with recent arguments that reliable algorithmic decision-making must be grounded in causal reasoning to ensure valid and trustworthy deployment (cf. Kern et al., 2025).

*Societal and fairness considerations:* While our method is model-agnostic and does not impose fairness constraints by design, it can be combined with fairness-aware modeling or post-hoc policy adjustment techniques. We encourage practitioners to monitor for disparate impact across subpopulations when deploying systems trained with DR$Q$-learner, especially in high-stakes domains.

## REPRODUCIBILITY STATEMENT

We have taken multiple steps to ensure the reproducibility of our results. All theoretical contributions, including identifiability results, the derivation of the efficient influence function, and the proof of double robustness and quasi-oracle efficiency, are presented in full detail in the main text and rigorously proven in Appendix D. Our algorithm is specified formally in Section 5.1 and summarized in pseudocode to facilitate implementation in Section 5.2. Hyperparameters, model classes, and training details are provided in Appendix E. For our empirical evaluation, we rely exclusively on environments from the OpenAI Gym package, as described in Section 6, which ensures that experiments can be exactly reproduced by other researchers. We also provide an anonymized, open-source implementation of DR$Q$-learner and scripts to reproduce all figures and tables, available at `https://github.com/EmilJavurek/Orthogonal-Q-in-MDPs`. Upon acceptance, we will make our code publicly available via GitHub repository. Together, these materials enable independent researchers to fully replicate both the theoretical and empirical results presented in this work.

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

## A   EXTENDED RELATED WORK

Here, we provide an extended related work to offer additional context for our work.

**Off-policy $Q$-function evaluation:** Methods targeting the off-policy $Q$-function from MDPs, as is our goal, are often presented as plug-in off-policy evaluation (OPE) methods. In the OPE literature, $Q_{\pi_e}$ is a nuisance function[12], where a new fitting procedure for $Q_{\pi_e}$ is taken to imply a new plug-in learner for OPE. An odd consequence of this is that the performance of $\hat{Q}_{\pi_e}$ *function* estimation is often evaluated only via the implied performance of the estimated *scalar average* off-policy policy value. Yet, many practical applications have direct interest in estimating individualized outcomes such as $Q_{\pi_e}$ to personalize medical decisions (Feuerriegel et al., 2024), and, hence, we focus here on estimating $Q_{\pi_e}$ directly. Existing $Q_{\pi_e}$ estimation techniques address the off-policy nature of the problem either explicitly via an inverse-propensity-weighting-like nuisance (Liu et al., 2018; Farajtabar et al., 2018; Uehara et al., 2020; Munos et al., 2016), or implicitly in the (supervised learning) target construction (Le et al., 2019; Lagoudakis & Parr, 2003; Precup et al., 2000; Harutyunyan et al., 2016). Finally, we mention the work of van der Laan et al. (2025), who have developed a debiased estimator for *linear functionals of* $Q_{\pi_e}$. While this generalizes debiased estimation from just OPE to all linear functionals of $Q_{\pi_e}$, it cannot be applied to $Q_{\pi_e}$ itself.

**Potential outcomes in MDPs:** Off-policy (potential outcome) estimation in MDPs is commonly encountered in OPE for RL. Here, the goal is to estimate the *scalar* policy value of an evaluation policy different from the one that generated the observed MDP trajectories. Various doubly-robust meta-learning methods have been developed to make the OPE estimate robust to errors in the learned nuisances (Kallus & Uehara, 2022; Farajtabar et al., 2018; Shi et al., 2021). Notably, Kallus & Uehara (2022) have derived the efficient influence function of the off-policy policy value and a corresponding efficient DR-learner. For a detailed statistical overview of OPE in RL, see Uehara et al. (2022). However, *none* of these learners are targeted at $Q$-function estimation, but only target the scalar policy value instead.

**Individualized potential outcomes over time:** Several methods have been proposed for estimating individualized potential outcomes in time-series settings (Lim et al., 2018; Bica et al., 2020; Melnychuk et al., 2022; Li et al., 2021; Hess et al., 2024; Lewis & Syrgkanis, 2021; Hess et al., 2026). These can be grouped into both model-based (e.g., adaptations of the transformer architecture for estimating individualized potential outcomes over time, such as in (Melnychuk et al., 2022)) and meta-learners (e.g., model-agnostic "recipes" for leveraging existing models to perform valid causal inference). Notably, Frauen et al. (2025) have derived a DR-learner and variations thereof. Methods in this stream target the conditional average potential outcome $Y_{t+\tau}, \tau > 0$ provided the entire history $H_t$ up to time $t$. Hence, while these methods could theoretically be adapted to target the long-term average of future outcomes (our goal), they do *not* take advantage of the Markov structure of MDPs and thus suffer from the curse of horizon.

**DTR:** An adjacent field are dynamic treatment regimes (DTR), which are concerned with optimizing (individualized) treatment assignment in a time-series setting. For an overview, see Chakraborty & Moodie (2013). While there are extensions of DTRs using machine learning (e.g., Theresa Blumlein et al., 2022), these have limitations for our setting. In particular, the DTR literature also typically does *not* consider the MDP setting, and methods from DTR thus suffer from the same curse of horizon as other general time-series methods.

---

[12]By $Q_{\pi_e}$, we mean the off-policy $Q$ of an evaluation policy $\pi_e$ that differs from the policy $\pi_b$ that we observe data from.

# B  ADDITIONAL DETAILS

**Identification.** To be able to estimate this causal quantity from observational data generated with $\pi_b$, we need the following standard identification assumptions (Robins et al., 2000; Uehara et al., 2022): (1) *Weak positivity:* The support of $\pi_e(\cdot \mid s)$ is included in the support of $\pi_b(\cdot \mid s)$ for any $s \in \mathcal{S}$. (2) *Consistency:* $R_t = R_t[A_t]$, almost surely. (3) *Unconfoundedness:* For any $a \in \mathcal{A}$, $A$ and $R[a]$ are conditionally independent given $S$, i.e., $A_t \perp\!\!\!\perp R_t[a_t] \mid S_t$. Of note, these assumptions are standard in the causal inference literature (Lim et al., 2018; Bica et al., 2020; Melnychuk et al., 2022; Seedat et al., 2022; Frauen et al., 2025)

**Curse of horizon.** The curse of horizon refers to the phenomenon that estimation error in off-policy RL grows exponentially with the time horizon when the estimand depends on long sequences of actions. The reason for this is the exponential decay in overlap between trajectories from the observed and evaluation policies. This is an inherent difficulty of the setting, for all estimation tasks in the off-policy setting and any corresponding approaches. The technical challenge lies in successfully and also efficiently leveraging the time-invariant Markov property of the MDP setting to break this curse. We refer to Kallus & Uehara (2022) for exhaustive treatment of this problem for off policy policy value estimation.

**Practical aspects of nuisance estimation.** The nuisances required by our second-stage model can in practice be quite complex and difficult to estimate. Here, we give several thoughts on this topic:

i) *The second stage is agnostic to choice of nuisance estimation model.* Apart from the sufficient convergence guarantees, the practitioner is free to employ any method of their choosing to estimate the nuisance. This is especially important in cases such as ours, where some of the nuisances are more complex.

ii) *The two-stage learner can, in principle, only ever improve upon the 1st stage.* Since our method requires the estimation of the target $Q_{\pi_e}$ in the first stage nuisance estimation, we are, of course, always free to stick with the nuisance estimate. Our DRQ learner is of use in settings where additional estimation complexity is worth it for the strong theoretical guarantees we provide in return. This motivation is natural for many high-stakes real-world applications such as medical applications listed in the Introduction.

iii) *The nuisances are only as complex as the underlying setting.* Given the principled derivation of our method with the efficient influence function, all nuisance functions present in the loss come from the statistical model inherent to the problem setting. It is worth pointing out existing papers that propose estimation methods for similar stationary density ratio nuisances such as ours ( e.g., the work in [1]). Conversely, methods not using these complex nuisances may be deceptively simple: FQE does not use this nuisance but is well known to have unpredictable failure regimes (see "deadly triad" problem, Sutton & Barto (2018)).

## C    Background on influence functions, orthogonal learning

In this section, we provide a brief overview of efficient influence functions and orthogonal learning, following the treatment in (Kennedy, 2022).

**Efficient influence function (EIF).** In semiparametric statistics, estimation is framed in terms of a statistical model $\{P \in \mathcal{P}\}$, where $\mathcal{P}$ denotes a family of probability distributions. We are interested in a functional $\psi : \mathcal{P} \to \mathbb{R}$. For instance, one might consider $\psi(P) = \mathbb{E}_P[R|S = s]$. If $\psi$ is sufficiently smooth, it admits a von Mises (distributional Taylor) expansion:

$$\psi(\bar{P}) - \psi(P) = \int \phi(t, \bar{P}) \, \mathrm{d}(\bar{P} - P)(t) + R_2(\bar{P}, P), \tag{18}$$

where $R_2(\bar{P}, P)$ is a second-order remainder term and $\phi(t, P)$ is the *efficient influence function* (EIF) of $\psi$. By definition, the EIF satisfies $\int \phi(t, P) \, \mathrm{d}P(t) = 0$ and $\int \phi(t, P)^2 \, \mathrm{d}P(t) < \infty$.

**Plug-in bias and bias correction.** Consider an estimator $\hat{P}$ of $P$ and the associated plug-in estimator $\psi(\hat{P})$. The expansion above implies a first-order *plug-in bias*:

$$\psi(\hat{P}) - \psi(P) = -\int \phi(t, \hat{P}) \, \mathrm{d}P(t) + R_2(\hat{P}, P), \tag{19}$$

because $\int \phi(t, \hat{P}) d\hat{P}(t) = 0$. Intuitively, simply plugging estimated nuisance functions into the identification formula generally leads to a biased estimator. A classical way to correct this bias is to estimate the term on the right-hand side and add it back, yielding a *one-step bias-corrected estimator*:

$$\hat{\psi} = \psi(\hat{P}) + \mathbb{P}_n\big[\phi(T, \hat{P})\big]. \tag{20}$$

This correction removes the leading-order bias, leaving only a second-order remainder.

**Debiased target loss and orthogonality.** While one-step correction works well for finite-dimensional parameters such as average treatment effects, it is not directly applicable for infinite-dimensional targets such as conditional treatment effects $\tau_t(X)$. In such settings, the EIF can still be used to construct a *debiased loss function* rather than directly de-biasing the target parameter. Minimizing this orthogonalized loss leads to estimators that are first-order insensitive to nuisance estimation error, which is the core idea behind Neyman-orthogonal learners.

# D PROOFS

## D.1 DERIVATION OF OUR LOSS

We construct our Neyman-orthogonal loss by debiasing the ERM *estimate* using the efficient influence function (EIF). We begin by taking the EIF of a standard MSE loss $L_{\pi_e}^1$.

STATISTICAL MODEL

First, we must define our statistical model. Let us define a model for observations $O = (S, A, R, \tilde{S}) \in \mathcal{S}^2 \times \mathcal{A} \times \mathcal{R}$ via

$$\mathcal{M} = \{p \mid p(o) = p(s)p(a|s)p(r|s,a)p(\tilde{s}|s,a); p(s)p(a|s) > 0\}. \tag{21}$$

We denote the (unknown) true data-generating (observational) distribution $\mathbb{P} \in \mathcal{M}$ and a one-dimensional parametrized submodel of distributions by

$$\mathcal{P}_\epsilon = \{p_\epsilon \mid p_\epsilon(o) = p(o) + \epsilon(p'(o) - p(o)); \epsilon \in [0,1)\} \tag{22}$$

where $\mathcal{P}_\epsilon \subset \mathcal{M}$, i.e., $p_\epsilon \in \mathcal{M}$. We take $p$ without subscript to be a density corresponding to $\mathbb{P}$, i.e., $p(a|s) = \pi_b(a|s), p(r|s,a) = p_r(r|s,a), p(\tilde{s}|s,a) = p_s(\tilde{s}|s,a)$.

We take the strategy advocated by Kennedy (2022), where, by cleverly choosing the parametric submodel to represent point-mass deviation from $\mathbb{P}$, i.e. the Dirac delta at point $O'$, $p'(o) = \delta(O' = o)$, the EIF derivation reduces to taking a Gateaux derivative

$$\mathbb{IF}(F(\mathbb{P}), O') = \frac{\partial}{\partial \epsilon} F(\mathbb{P}_\epsilon)\Big|_{\epsilon=0}. \tag{23}$$

We refer to Kennedy (2022); Fisher & Kennedy (2018) for comprehensive tutorials and technical details of efficient influence functions.

TAKING THE EIF OF $L_{\pi_e}^1$

With the MSE population risk under the evaluation distribution $L_{\pi_e}^1$ defined as

$$L_{\pi_e}^1(\eta, g) = \mathbb{E}_{O \sim p_e}\left[(Q_{\pi_e}(S,a) - g(S,a))^2\right] = \mathbb{E}_{O \sim p_b}\left[\sum_a \pi_e(a|S)(Q_{\pi_e}(S,a) - g(S,a))^2\right], \tag{24}$$

we take the EIF via

$$\mathbb{IF}(L_{\pi_e}^1(\eta, g), O') = \tag{25}$$

$$= \sum_a \pi_e(a|S')(Q_{\pi_e}(S,a) - g(S,a))^2 - L_{\pi_e}^1(\eta, g) \tag{26}$$

$$+ \int \sum_a p_b(s)\pi_e(a|s)2(Q_{\pi_e}(s,a) - g(s,a))\mathbb{IF}(Q_{\pi_e}(s,a), O')\mathrm{ds}. \tag{27}$$

To derive $\mathbb{IF}(Q_{\pi_e}(s,a), O')$, we decompose the $Q_{\pi_e}$ via its definition $Q_{\pi_e}(s,a) = \mathbb{E}_\pi\left[\sum_{t=0}^\infty \gamma^t R_t | S_0 = s, A_0 = a\right]$. Taking the EIF of the individual elements of the sum, we have, sequentially, the EIF for the the null and first conditional expected reward by

$$\mathbb{IF}\left(\mathbb{E}_{\pi_e}[R_0|S_0 = s_0, A_0 = a_0], O'\right) = \frac{\delta(s_0 = S', a_0 = A')}{p_b(S = s_0, A = a_0)}(R' - \mathbb{E}[R_0|S_0 = s_0, A_0 = a_0]) \tag{28}$$

$$\mathbb{IF}\left(\mathbb{E}_{\pi_e}[R_1|S_0 = s_0, A_0 = a_0], O'\right) = \tag{29}$$

$$= \mathbb{IF}\left(\int p(s_1|s_0, a_0)\pi_e(a_1|s_1)p(r_1|s_1, a_1)r_1\mathrm{ds}_1\mathrm{da}_1\mathrm{dr}_1\right) \tag{30}$$

$$= \int \left\{\frac{\delta(s_0 = S', a_0 = A')}{p_b(S = s_0, A = a_0)}(\delta(s_1 = \tilde{S}') - p(s_1|s_0, a_0))\right\}\pi_e(a_1|s_1)p(r_1|s_1, a_1)r_1\mathrm{ds}_1\mathrm{da}_1\mathrm{dr}_1 \tag{31}$$

$$+ \int p(s_1|s_0, a_0)\pi_e(a_1|s_1)\left\{\frac{\delta(s_1 = S', a_1 = A')}{p_b(S = s_1, A = a_1)}(\delta(r_1 = R') - p(r_1|s_1, a_1))\right\}r_1\mathrm{ds}_1\mathrm{da}_1\mathrm{dr}_1 \tag{32}$$

$$= \frac{\delta(s_0 = S', a_0 = A')}{p_b(S = s_0, A = a_0)}\left\{\mathbb{E}_{\pi_e}[R_1|S_1 = \tilde{S}'] - \mathbb{E}_{\pi_e}[R_1|S_0 = s_0, A_0 = a_0]\right\} \tag{33}$$

$$+ \frac{p_e(S_1 = S', A_1 = A'|S_0 = s_0, A_0 = a_0)}{p_b(S = S', A = A')}\left\{R' - \mathbb{E}_{\pi_e}[R_1|S_1 = S', A_1 = A']\right\}. \tag{34}$$

We further yield the EIF for the second conditional expected reward by

$$\mathbb{IF}\left(\mathbb{E}_{\pi_e}[R_2|S_0 = s_0, A_0 = a_0], O'\right) = \tag{35}$$

$$= \mathbb{IF}\left(\int p(s_1|s_0, a_0)\pi_e(a_1|s_1)p(s_2|s_1, a_1)\pi_e(a_2|s_2)p(r_2|s_2, a_2)r_2\mathrm{d}s_1\mathrm{d}a_1\mathrm{d}s_2\mathrm{d}a_2\mathrm{d}r_2\right) \tag{36}$$

$$= \int \left\{\frac{\delta(s_0 = S', a_0 = A')}{p_b(S = s_0, A = a_0)}(\delta(s_1 = \tilde{S}') - p(s_1|s_0, a_0))\right\}\pi_e(a_1|s_1)p(s_2|s_1, a_1)\pi_e(a_2|s_2)p(r_2|s_2, a_2)r_2\mathrm{d}s_1\mathrm{d}a_1\mathrm{d}s_2\mathrm{d}a_2\mathrm{d}r_2 \tag{37}$$

$$+ \int p(s_1|s_0, a_0)\pi_e(a_1|s_1)\left\{\frac{\delta(s_1 = S', a_1 = A')}{p_b(S = s_1, A = a_1)}(\delta(s_2 = \tilde{S}') - p(s_2|s_1, a_1)\right\}\pi_e(a_2|s_2)p(r_2|s_2, a_2)r_2\mathrm{d}s_1\mathrm{d}a_1\mathrm{d}s_2\mathrm{d}a_2\mathrm{d}r_2 \tag{38}$$

$$+ \int p(s_1|s_0, a_0)\pi_e(a_1|s_1)p(s_2|s_1, a_1)\pi_e(a_2|s_2)\left\{\frac{\delta(s_2 = S', a_2 = A')}{p_b(S = s_2, A = a_2)}(\delta(r_2 = R') - p(r_2|s_2, a_2))\right\}r_2\mathrm{d}s_1\mathrm{d}a_1\mathrm{d}s_2\mathrm{d}a_2\mathrm{d}r_2 \tag{39}$$

$$= \frac{\delta(s_0 = S', a_0 = A')}{p_b(S = s_0, A = a_0)}\left\{\mathbb{E}_{\pi_e}[R_2|S_1 = \tilde{S}'] - \mathbb{E}_{\pi_e}[R_2|S_0 = s_0, A_0 = a_0]\right\} \tag{40}$$

$$+ \frac{p_e(S_1 = S', A_1 = A'|S_0 = s_0, A_0 = a_0)}{p_b(S = S', A = A')}\left\{\mathbb{E}_{\pi_e}[R_2|S_2 = \tilde{S}'] - \mathbb{E}_{\pi_e}[R_2|S_1 = S', A_1 = A']\right\} \tag{41}$$

$$+ \frac{p_e(S_2 = S', A_2 = A'|S_0 = s_0, A_0 = a_0)}{p_b(S = S', A = A')}\left\{R' - \mathbb{E}_{\pi_e}[R_2|S_2 = S', A_2 = A']\right\}. \tag{42}$$

Generally, for $k \geq 1$ (where we abuse the notation with the arrows for readability), we thus have

$$\mathbb{IF}\left(\mathbb{E}_{\pi_e}[R_k|S_0 = s_0, A_0 = a_0], O'\right) =$$

$$= \mathbb{IF}\left(\int p_e(s_1 \to r_k|s_0, a_0)r_k\mathrm{d}s_1 \to \mathrm{dr}_k, O'\right)$$

$$= \int \sum_{t=1}^{k}\mathbb{IF}(p(s_t|s_{t-1}, a_{t-1}))\frac{p_e(s_1 \to r_k|s_0, a_0)}{p(s_t|s_{t-1}, a_{t-1})}r_k\mathrm{d}s_1 \to \mathrm{dr}_k$$

$$+ \int \mathbb{IF}(p(r_k|s_k, a_k))\frac{p_e(s_1 \to r_k|s_0, a_0)}{p(r_k|s_k, a_k)}r_k\mathrm{d}s_1 \to \mathrm{dr}_k$$

$$= \frac{\delta(s_0 = S', a_0 = A')}{p_b(S = s_0, A = a_0)}\left\{\mathbb{E}_{\pi_e}[R_k|S_1 = \tilde{S}'] - \mathbb{E}_{\pi_e}[R_k|S_0 = s_0, A_0 = a_0]\right\}$$

$$+ \sum_{t=1}^{k-1}\frac{p_e(S_t = S', A_t = A'|S_0 = s_0, A_0 = a_0)}{p_b(S = S', A = A')}\left\{\mathbb{E}_{\pi_e}[R_k|S_{t+1} = \tilde{S}'] - \mathbb{E}_{\pi_e}[R_k|S_t = S', A_t = A']\right\}$$

$$+ \frac{p_e(S_k = S', A_k = A'|S_0 = s_0, A_0 = a_0)}{p_b(S_k = S', A_k = A')}\left\{R' - \mathbb{E}_{\pi_e}[R_k|S_k = S', A_k = A']\right\}.$$

Putting it together, we get

$$\mathbb{IF}\left(\mathbb{E}_{\pi_e}[\sum_{t=0}^{k}\gamma^t R_t|S_0 = s_0, A_0 = a_0], O'\right)$$

$$= \sum_{t=0}^{k}\gamma^t\mathbb{IF}\left(\mathbb{E}_{\pi_e}[R_t|S_0 = s_0, A_0 = a_0]\right)$$

$$= \frac{\delta(s_0 = S', a_0 = A')}{p_b(S = s_0, A = a_0)}\left\{R' + \sum_{t=1}^{k}\gamma^t\mathbb{E}_{\pi_e}[R_t|S_1 = \tilde{S}'] - \sum_{t=0}^{k}\gamma^t\mathbb{E}_{\pi_e}[R_t|S_0 = s_0, A_0 = a_0]\right\}$$

$$+ \frac{\gamma p_e(S_1 = S', A_1 = A'|S_0 = s_0, A_0 = a_0)}{p_b(S = S', A = A')}\left\{R' + \sum_{t=2}^{k}\gamma^{t-1}\mathbb{E}_{\pi_e}[R_t|S_2 = \tilde{S}'] - \sum_{t=1}^{k}\gamma^{t-1}\mathbb{E}_{\pi_e}[R_t|S_1 = S', A_1 = A']\right\}$$

$$\vdots$$

$$+ \frac{\gamma^j p_e(S_j = S', A_j = A'|S_0 = s_0, A_0 = a_0)}{p_b(S = S', A = A')}\left\{R' + \sum_{t=j+1}^{k}\gamma^{t-j}\mathbb{E}_{\pi_e}[R_t|S_{j+1} = \tilde{S}'] - \sum_{t=j}^{k}\gamma^{t-j}\mathbb{E}_{\pi_e}[R_t|S_j = S', A_j = A']\right\}$$

$$\vdots$$

$$+ \frac{\gamma^k p_e(S_k = S', A_k = A'|S_0 = s_0, A_0 = a_0)}{p_b(S = S', A = A')}\left\{R' - \mathbb{E}_{\pi_e}[R_k|S_k = S', A_k = A']\right\},$$

for $2 \le j < k$. Now, we recognize that, for all second terms in the brackets, we yield

$$\sum_{t=j+1}^{k} \gamma^{t-j} \mathbb{E}_{\pi_e}[R_t | S_{j+1} = \tilde{S}'] = \sum_{t=0}^{k-(j+1)} \gamma^{t+1} \mathbb{E}_{\pi_e}[R_t | S_0 = \tilde{S}'] \to \gamma v_{\pi_e}(\tilde{S}') \text{ as } k \to \infty, \quad (43)$$

and, analogously, for the final terms, we yield

$$\sum_{t=j}^{k} \gamma^{t-j} \mathbb{E}_{\pi_e}[R_t | S_j = S', A_j = A'] = \sum_{t=0}^{k-j} \gamma^{t} \mathbb{E}_{\pi_e}[R_t | S_0 = S', A_0 = A'] \to Q_{\pi_e}(S', A') \text{ as } k \to \infty. \quad (44)$$

Recognizing that, in the limit, the brackets are equivalent, we find the limit of the whole expression to be

$$\mathbb{IF}(Q_{\pi_e}(s_0, a_0), O') = \mathbb{IF}\left( \lim_{k \to \infty} \mathbb{E}_{\pi_e}[\sum_{t=0}^{k} \gamma^t R_t | S_0 = s_0, A_0 = a_0], O' \right) \quad (45)$$

$$= \left( \frac{\delta(s_0 = S', a_0 = A')}{p_b(S')\pi_b(A'|S')} + \frac{\pi_e(A'|S')}{\pi_b(A'|S')} w_{e/b}(S'|s_0, a_0) \right) \left\{ R' + \gamma v_{\pi_e}(\tilde{S}') - Q_{\pi_e}(S', A') \right\}. \quad (46)$$

Plugging the result into the EIF of $L^1_{\pi_e}$, we obtain

$$\mathbb{IF}(L^1_{\pi_e}(\eta, g), O') = \quad (47)$$

$$= \sum_a \pi_e(a|S') \left( Q_{\pi_e}(S', a) - g(S', a) \right)^2 - L^1_{\pi_e}(\eta, g) + 2 \left\{ R' + \gamma v_{\pi_e}(\tilde{S}') - Q_{\pi_e}(S', A') \right\} \frac{\pi_e(A'|S')}{\pi_b(A'|S')} \quad (48)$$

$$\times \left[ Q_{\pi_e}(S', A') - g(S', A') + \mathbb{E}_{s,a \sim p_b(s)\pi_e(a|s)} \left[ (Q_{\pi_e}(s, a) - g(s, a)) w_{e/b}(S'|s, a) \right] \right]. \quad (49)$$

DEBIASING THE $L^1_{\pi_e}$

Applying the EIF to debias the ERM *estimate* of the population risk, we obtain a debiased loss

$$\hat{L}^2_{\pi_e}(\eta, g) = \hat{\mathbb{E}}_{O' \sim p_b} \left[ L^1_{\pi_e}(\eta, g) + \mathbb{IF}(L^1_{\pi_e}(\eta, g), O') \right] \quad (50)$$

$$= \hat{\mathbb{E}}_{O' \sim p_b} \left\{ \sum_a \pi_e(a|S') \left( Q_{\pi_e}(S', a) - g(S', a) \right)^2 + 2 \left\{ R' + \gamma v_{\pi_e}(\tilde{S}') - Q_{\pi_e}(S', A') \right\} \frac{\pi_e(A'|S')}{\pi_b(A'|S')} \quad (51)$$

$$\times \left[ Q_{\pi_e}(S', A') - g(S', A') + \mathbb{E}_{s,a \sim p_b(s)\pi_e(a|s)} \left[ (Q_{\pi_e}(s, a) - g(s, a)) w_{e/b}(S'|s, a) \right] \right] \right\}. \quad (52)$$

We complete the squares to obtain a final loss:

$$\hat{L}^2_{\pi_e}(\eta, g) \overset{\arg\min}{=} \hat{L}^3_{\pi_e}(\eta, g) \quad (53)$$

$$= \hat{\mathbb{E}}_{O' \sim p_b} \left[ \sum_a \pi_e(a|S') \left( 2\frac{\delta(A' = a)}{\pi_b(A'|S')} \left\{ R' + \gamma v_{\pi_e}(\tilde{S}') - Q_{\pi_e}(S', A') \right\} + Q_{\pi_e}(S', a) - g(S', a) \right)^2 \right] \quad (54)$$

$$+ \hat{\mathbb{E}}_{O' \sim p_b, s \sim p_b(s)} \left[ \sum_a \pi_e(a|s) \left( 2\frac{\pi_e(A'|S')}{\pi_b(A'|S')} w_{e/b}(S'|s, a) \left\{ R' + \gamma v_{\pi_e}(\tilde{S}') - Q_{\pi_e}(S', A') \right\} + Q_{\pi_e}(s, a) - g(s, a) \right)^2 \right] \quad (55)$$

This completes the derivation. The corresponding proof that $L^3_{\pi_e}$ is minimized by $Q_{\pi_e}$ can be found in Appendix D.5. We continue with the proof that $L^3_{\pi_e}$ is Neyman-orthogonal.

### D.1.1 INTUITION BEHIND $\phi_1$ AND $\phi_2$

Both $\phi_1$ and $\phi_2$ are the respective targets or "pseudo-outcomes" of the MSE subcomponents of the Neyman-oOrthogonal loss. Each contains the $Q_{\pi_e}$ term, which, if alone, would correspond simply to the standard MSE loss without debiasing. This is the non-Neyman-orthogonal starting point that we aim to debias in order to obtain robustness wrt. Nuisance estimation error. The additional debiasing terms of both $\phi_1$ and $\phi_2$ are two variations of temporal difference error (curly brackets $R' + \gamma v_{\pi_e}(\tilde{S}') - Q_{\pi_e}(S', A')$) scaled by an importance-sampling-like density ratio.

While the density ratios here are quite complicated, the overall form of the Neyman-orthogonal loss is not. The debiasing being of the form "mean zero error scaled by density ratio" is common across

many instances of DR learners in standard causal inference, including but not limited to ATE and CATE estimation.

In $\phi_1$, the density-ratio is the simple one-step (first-step) inverse propensity weighting. In $\phi_2$, the density-ratio is of the conditional stationary state density. Borrowing from Markov chain terminology, $2\frac{\pi_e(A'|S')}{\pi_b(A'|S')}w_{e/b}(S' \mid s, a)$ is the ratio of the likelihood of observing the state-action pair $(S', A')$ following from the stationary distribution of the chain induced by following policy $\pi_e$ (conditional on the chain having begun at pair $s, a$) versus following from the initial distribution following policy $\pi_b$.

### D.2  Neyman-orthogonality of $L^3$

First, we state a useful Lemma:

**Lemma 3** (Expected TD error is zero). *The expectation of the temporal difference error of $\pi_e$ w.r.t. to any measurable distribution in the model (i.e., the distribution generated by any policy $\pi$), weighted by any (measurable and bounded) function $f(S', A')$ is zero.*

$$\mathbb{E}_\pi\left[f(S', A')\left(R' + \gamma v_{\pi_e}(\tilde{S}') - Q_{\pi_e}(S', A')\right)\right] = 0 \tag{56}$$

*Proof.*

$$\mathbb{E}_\pi\left[f(S', A')\left(R' + \gamma v_{\pi_e}(\tilde{S}') - Q_{\pi_e}(S', A')\right)\right] = \tag{57}$$

$$=\mathbb{E}_\pi\left[\mathbb{E}_\pi\left[f(S', A')\left(R' + \gamma v_{\pi_e}(\tilde{S}') - Q_{\pi_e}(S', A')\right) \mid S', A'\right]\right] \tag{58}$$

$$=\mathbb{E}_\pi\left[f(S', A')\left(\mathbb{E}_\pi\left[R' + \gamma v_{\pi_e}(\tilde{S}') \mid S', A'\right] - Q_{\pi_e}(S', A')\right)\right] \tag{59}$$

$$=\mathbb{E}_\pi\left[f(S', A')\left(Q_{\pi_e}(S', A') - Q_{\pi_e}(S', A')\right)\right] = 0 \tag{60}$$

$\square$

PROOF OF NEYMAN-ORTHOGONALITY

*Proof.* We show the Neyman-orthogonality of our loss $L^3_{\pi_e}$. We define

$$\Delta\hat{g}(\cdot) \triangleq \hat{g}(\cdot) - g^*(\cdot). \tag{61}$$

The first (Gateaux) derivative is

$$D_g L^3_{\pi_e}(\eta, g^*)[\hat{g} - g^*] = \tag{62}$$

$$= -2\mathbb{E}_{O'\sim p_b, a\sim\pi_e(a|S')}\left[\Delta\hat{g}(S', a)\left(2\frac{\delta(A' = a)}{\pi_b(A'|S')}\left\{R' + \gamma v_{\pi_e}(\tilde{S}') - Q_{\pi_e}(S', A')\right\} + Q_{\pi_e}(S', a) - g^*(S', a)\right)\right] \tag{63}$$

$$- 2\mathbb{E}_{O'\sim p_b; s, a\sim p_b(s)\pi_e(a|s)}\left[\Delta\hat{g}(s, a)\left(2\frac{\pi_e(A'|S')}{\pi_b(A'|S')}w_{e/b}(S'|s, a)\left\{R' + \gamma v_{\pi_e}(\tilde{S}') - Q_{\pi_e}(S', A')\right\} + Q_{\pi_e}(s, a) - g^*(s, a)\right)\right] \tag{64}$$

Continuing, we take second derivatives with respect to all the nuisances $\eta = (\pi_b, w_{e/b}, Q_{\pi_e})$. First, for $\pi_b$ we yield

$$D_{\pi_b} D_g L^3_{\pi_e}(\eta, g^*)[\hat{g} - g^*, \hat{\pi}_b - \pi_b] \tag{65}$$

$$= -2\mathbb{E}_{O'\sim p_b, a\sim\pi_e(a|S')}\left[\Delta\hat{g}(S', a)\Delta\hat{\pi}_b(A'|S')2\delta(A' = a)\left\{R' + \gamma v_{\pi_e}(\tilde{S}') - Q_{\pi_e}(S', A')\right\}(-1)\frac{1}{\pi_b(A'|S')^2}\right] \tag{66}$$

$$- 2\mathbb{E}_{O'\sim p_b; s, a\sim p_b(s)\pi_e(a|s)}\left[\Delta\hat{g}(s, a)\Delta\hat{\pi}_b(A'|S')2\pi_e(A'|S')w_{e/b}(S'|s, a)\left\{R' + \gamma v_{\pi_e}(\tilde{S}') - Q_{\pi_e}(S', A')\right\}(-1)\frac{1}{\pi_b(A'|S')^2}\right] \tag{67}$$

$$=0. \tag{68}$$

We use Lemma 3 to show equality to zero.
Second, for $w_{e/b}$, we yield

$$D_{w_{e/b}} D_g L_{\pi_e}^3(\eta, g^*)[\hat{g} - g^*, \hat{w}_{e/b} - w_{e/b}] = \tag{69}$$

$$= -2\mathbb{E}_{O' \sim p_b; s, a \sim p_b(s)\pi_e(a|s)}\left[\Delta\hat{g}(s,a)2\frac{\pi_e(A'|S')}{\pi_b(A'|S')}\left\{R' + \gamma v_{\pi_e}(\tilde{S}') - Q_{\pi_e}(S', A')\right\}\Delta\hat{w}_{e/b}(S'|s,a)\right] \tag{70}$$

$$= 0 \tag{71}$$

Lastly, for $Q_{\pi_e}$, we have

$$D_{Q_{\pi_e}} D_g L_{\pi_e}^3(\eta, g^*)[\hat{g} - g^*, \hat{Q}_{\pi_e} - Q_{\pi_e}] = \tag{72}$$

$$= -2\mathbb{E}_{O' \sim p_b, a \sim \pi_e(a|S')}\left[\Delta\hat{g}(S', a)\left(2\frac{\delta(A'=a)}{\pi_b(A'|S')}\left(\gamma\mathbb{E}_{\tilde{A}' \sim \pi_e(\tilde{A}'|\tilde{S}')}[\Delta\hat{Q}_{\pi_e}(\tilde{S}', \tilde{A}')] - \Delta\hat{Q}_{\pi_e}(S', A')\right) + \Delta\hat{Q}_{\pi_e}(S', a)\right)\right] \tag{73}$$

$$-2\mathbb{E}_{O' \sim p_b; s, a \sim p_b(s)\pi_e(a|s)}\left[2\frac{\pi_e(A'|S')}{\pi_b(A'|S')}w_{e/b}(S'|s,a)\Delta\hat{g}(s,a)\left(\gamma\mathbb{E}_{\tilde{A}' \sim \pi_e(\tilde{A}'|\tilde{S}')}[\Delta\hat{Q}_{\pi_e}(\tilde{S}', \tilde{A}')] - \Delta\hat{Q}_{\pi_e}(S', A')\right)\right. \tag{74}$$

$$\left. + \Delta\hat{g}(s,a)\Delta\hat{Q}_{\pi_e}(s,a)\right] \tag{75}$$

$$= -4\gamma\mathbb{E}_{O' \sim p_b, a \sim \pi_e(a|S')}\left[\Delta\hat{g}(S', a)\frac{\delta(A'=a)}{\pi_b(A'|S')}\mathbb{E}_{\tilde{A}' \sim \pi_e(\tilde{A}'|\tilde{S}')}[\Delta\hat{Q}_{\pi_e}(\tilde{S}', \tilde{A}')]\right] \tag{76}$$

$$-4\mathbb{E}_{O' \sim p_b; s, a \sim p_b(s)\pi_e(a|s)}\left[\frac{\pi_e(A'|S')}{\pi_b(A'|S')}w_{e/b}(S'|s,a)\Delta\hat{g}(s,a)\left(\gamma\mathbb{E}_{\tilde{A}' \sim \pi_e(\tilde{A}'|\tilde{S}')}[\Delta\hat{Q}_{\pi_e}(\tilde{S}', \tilde{A}')] - \Delta\hat{Q}_{\pi_e}(S', A')\right)\right] \tag{77}$$

$$= -4\mathbb{E}_{\substack{s \sim p_b(s), a \sim \pi_e(a|s), \tilde{s} \sim p(\tilde{s}|s,a), \tilde{a} \sim \pi_e(\tilde{a}|\tilde{s}) \\ S' \sim \beta_e(S'|s,a), A' \sim \pi_e(A'|S'), \tilde{S}' \sim p(\tilde{S}'|S',A'), \tilde{A}' \sim \pi_e(\tilde{A}'|\tilde{S}')}}\left[\Delta\hat{g}(s,a)\left(\gamma\Delta\hat{Q}_{\pi_e}(\tilde{s}, \tilde{a}) + \gamma\Delta\hat{Q}_{\pi_e}(\tilde{S}', \tilde{A}') - \Delta\hat{Q}_{\pi_e}(S', A')\right)\right] \tag{78}$$

$$= 0. \tag{79}$$

Since $\gamma(\tilde{s} + \tilde{S}') \overset{d}{=} S'$, the distribution of $S'$ in the final expectation is $\beta_e(S'|s,a) \triangleq \frac{1-\gamma}{\gamma}\sum_{t=1}^{\infty}\gamma^t p_e(S_t = S'|S_0 = s, A_0 = a)$, which can be interpreted as a conditional discounted stationary state distribution.

Hence, $L_{\pi_e}^3$ is Neyman-orthogonal. $\qquad\square$

### D.3 QUASI-ORACLE EFFICIENCY

We prove our loss achieves quasi-oracle efficiency. We write $L^3_{\pi_e}$ as

$$L^3_{\pi_e}(\eta, g) = \mathbb{E}_{O' \sim p_b; a \sim \pi_e(a|S')} \left[ (\phi_1 - g(S', a))^2 \right] + \mathbb{E}_{O' \sim p_b; s, a \sim p_b(s)\pi_e(a|s)} \left[ (\phi_2 - g(s, a))^2 \right], \quad (80)$$

where we define

$$\phi_1 = 2\frac{\delta(A' = a)}{\pi_b(A'|S')} \left\{ R' + \gamma v_{\pi_e}(\tilde{S}') - Q_{\pi_e}(S', A') \right\} + Q(S', a), \quad (81)$$

$$\phi_2 = 2\frac{\pi_e(A'|S')}{\pi_b(A'|S')} w_{e/b}(S'|s, a) \left\{ R' + \gamma v_{\pi_e}(\tilde{S}') - Q_{\pi_e}(S', A') \right\} + Q_{\pi_e}(s, a). \quad (82)$$

Additionally, we repeat the definitions $\hat{g} = \arg\min_{g \in \mathcal{G}} L^3_{\pi_e}(\hat{\eta}, g)$ and $g^* = \arg\min_{g \in \mathcal{G}} L^3_{\pi_e}(\eta, g)$, where $\hat{\eta}$ are the estimated nuisances and $\eta$ are the (unknown) true oracle nuisances.

So, we now arrive at

$$L^3_{\pi_e}(\hat{\eta}, \hat{g}) = \mathbb{E}_{O' \sim p_b; a \sim \pi_e(a|S')} \left[ (\hat{\phi}_1 - \hat{g}(S', a) + g^*(S', a) - g^*(S', a))^2 \right] \quad (83)$$

$$+ \mathbb{E}_{O' \sim p_b; s, a \sim p_b(s)\pi_e(a|s)} \left[ (\hat{\phi}_2 - \hat{g}(s, a) + g^*(s, a) - g^*(s, a))^2 \right] \quad (84)$$

$$= L^3_{\pi_e}(\hat{\eta}, g^*) + 2\mathbb{E}_{s, a \sim p_b(s)\pi_e(a|s)} \left[ (g^*(s, a) - \hat{g}(s, a))^2 \right] + D_g L^3_{\pi_e}(\hat{\eta}, g^*)[\Delta \hat{g}], \quad (85)$$

where we obtain the last line by decomposing the square and recognizing terms. Rearranging, we see

$$2\|g^* - \hat{g}\|^2_{2, p_b \pi_e} = R_g - D_g L^3_{\pi_e}(\hat{\eta}, g^*)[\Delta \hat{g}], \quad (86)$$

where $R_g = L^3_{\pi_e}(\hat{\eta}, \hat{g}) - L^3_{\pi_e}(\hat{\eta}, g^*)$.

We now arrange $D_g L^3_{\pi_e}(\hat{\eta}, g^*)$ via a second-order Taylor approximation to the true $\eta$, i.e.,

$$D_g L^3_{\pi_e}(\hat{\eta}, g^*)[\Delta \hat{g}] = D_g L^3_{\pi_e}(\eta, g^*)[\Delta \hat{g}] \quad (87)$$

$$+ D_\eta D_g L^3_{\pi_e}(\eta, g^*)[\Delta \hat{g}, \Delta \hat{\eta}] \quad (= 0 \text{ by Neyman-Orthogonality}) \quad (88)$$

$$+ \frac{1}{2} D_\eta^2 D_g L^3_{\pi_e}(\bar{\eta}, g^*)[\Delta \hat{g}, \Delta \hat{\eta}, \Delta \hat{\eta}], \quad (89)$$

for some $\bar{\eta} \in \text{star}(\mathcal{H}, \eta)$, where denotes the star-shaped set with respect to $\eta$. The last term is of the form

$$D_\eta^2 D_g L^3_{\pi_e}(\bar{\eta}, g^*)[\Delta \hat{g}, \Delta \hat{\eta}, \Delta \hat{\eta}] = \quad (90)$$

$$= -2\mathbb{E}_{O' \sim p_b; a \sim \pi_e(a|S')} \left[ \Delta \hat{g}(S', a) \Delta \hat{\eta}^\top \nabla_{\eta\eta} \bar{\phi}_1 \Delta \hat{\eta} \right] - 2\mathbb{E}_{O' \sim p_b; s, a \sim p_b(s)\pi_e(a|s)} \left[ \Delta \hat{g}(s, a) \Delta \hat{\eta}^\top \nabla_{\eta\eta} \bar{\phi}_2 \Delta \hat{\eta} \right]. \quad (91)$$

Continuing, we then have

$$2\|g^* - \hat{g}\|^2_{2, p_b \pi_e} = R_g - D_g L^3_{\pi_e}(\eta, g^*)[\Delta \hat{g}] - D_\eta^2 D_g L^3_{\pi_e}(\bar{\eta}, g^*)[\Delta \hat{g}, \Delta \hat{\eta}, \Delta \hat{\eta}] \quad (92)$$

$$\leq R_g - \frac{1}{2} D_\eta^2 D_g L^3_{\pi_e}(\bar{\eta}, g^*)[\Delta \hat{g}, \Delta \hat{\eta}, \Delta \hat{\eta}] \quad (93)$$

$$\leq R_g + \|g^* - \hat{g}\|_{p_b \pi_e} \left\{ \sum_{i=\{1,2,3\}; j=\{1,2,3\}; k=\{1,2\}} \sqrt{\mathbb{E}\left[ (\Delta \hat{\eta}_i [\nabla_{\eta\eta} \bar{\phi}_k]_{i,j} \Delta \hat{\eta}_j)^2 \right]} \right\} \quad (94)$$

$$\leq R_g + \|g^* - \hat{g}\|^2_{p_b \pi_e} \left( \sum_{i,j,k} \delta_{ijk} \right) + \left\{ \sum_{i=\{1,2,3\}; j=\{1,2,3\}; k=\{1,2\}} \frac{1}{\delta_{ijk}} \mathbb{E}\left[ (\Delta \hat{\eta}_i [\nabla_{\eta\eta} \bar{\phi}_k]_{i,j} \Delta \hat{\eta}_j)^2 \right] \right\}, \quad (95)$$

where we achieve the first inequality by recognizing $D_g L^3_{\pi_e}(\eta, g^*)[\Delta \hat{g}] \geq 0$, the second using the Cauchy-Schwarz inequality, and the third using the AM-GM inequality for any constants $\delta_{ijk} > 0$

such that $\sum_{i,j,k} \delta_{ijk} < 2$. This then finally results the inequality

$$2\|g^* - \hat{g}\|_{2,p_b\pi_e}^2 \leq \frac{1}{2 - \sum_{i,j,k} \delta_{ijk}} \left( R_g + \mathbb{E}\left[ C_1^2 \Delta^4 \hat{\pi}_b + C_2^2 \Delta^2 \hat{\pi}_b \Delta^2 \hat{Q}_{\pi_e} + C_3^2 \Delta^2 \hat{\pi}_b \Delta^2 \hat{w}_{e/b} + C_4^2 \Delta^2 \hat{w}_{e/b} \Delta^2 \hat{Q}_{\pi_e} \right] \right) \tag{96}$$

$$\leq \frac{1}{2 - \sum_{i,j,k} \delta_{ijk}} \left( R_g + \|C_1 \Delta^2 \hat{\pi}_b\|_2^2 + \|C_2 \Delta \hat{\pi}_b \Delta \hat{Q}_{\pi_e}\|_2^2 + \|C_3 \Delta \hat{\pi}_b \Delta \hat{w}_{e/b}\|_2^2 + \|C_4 \Delta \hat{w}_{e/b} \Delta \hat{Q}_{\pi_e}\|_2^2 \right) \tag{97}$$

$$\lesssim \frac{1}{2 - \sum_{i,j,k} \delta_{ijk}} \left( R_g + \|\Delta^4 \hat{\pi}_b\|_2^2 + \|\Delta^2 \hat{\pi}_b \Delta^2 \hat{Q}_{\pi_e}\|_2^2 + \|\Delta^2 \hat{\pi}_b \Delta^2 \hat{w}_{e/b}\|_2^2 + \|\Delta^2 \hat{w}_{e/b} \Delta^2 \hat{Q}_{\pi_e}\|_2^2 \right), \tag{98}$$

where the $C_1, \ldots, C_4$ collect all terms that do not contain $\Delta$ terms of difference between estimated and true nuisances. In the last steps, $x \lesssim y$ is taken to mean there exists a constant $M > 0$ s.t. $x \leq My$. The last inequality is achieved by extracting $\Delta\bar{\eta}$ terms from the $C$s and noting $\|\Delta\bar{\eta}\| \leq \|\Delta\hat{\eta}\|$ since $\bar{\eta}$ lies between $\hat{\eta}$ and the oracle $\eta$. For clarity of exposition, the computation of the Hessian terms through which the $C's$ contain $\Delta\bar{\eta}$ terms is postponed to the end of the proof. Lastly, we make use of Hölder's inequality

$$2\|g^* - \hat{g}\|_{2,p_b\pi_e}^2 \tag{99}$$

$$\lesssim \frac{1}{2 - \sum_{i,j,k} \delta_{ijk}} \left( R_g + \|\Delta^4 \hat{\pi}_b\|_2^2 + \|\Delta^2 \hat{\pi}_b\|_4^2 \|\Delta^2 \hat{Q}_{\pi_e}\|_4^2 + \|\Delta^2 \hat{\pi}_b\|_4^2 \|\Delta^2 \hat{w}_{e/b}\|_4^2 + \|\Delta^2 \hat{w}_{e/b}\|_4^2 \|\Delta^2 \hat{Q}_{\pi_e}\|_4^2 \right) \tag{100}$$

$$\lesssim \|\Delta^4 \hat{\pi}_b\|_2^2 + \|\Delta^2 \hat{\pi}_b \Delta^2 \hat{Q}_{\pi_e}\|_2^2 + \|\Delta^2 \hat{\pi}_b \Delta^2 \hat{w}_{e/b}\|_2^2 + \|\Delta^2 \hat{w}_{e/b} \Delta^2 \hat{Q}_{\pi_e}\|_2^2 \tag{101}$$

This finishes the proof. The double-robustness property is proved trivially by plugging in the condition (either $\Delta\hat{Q}_{\pi_e} \to 0$ or $\Delta\hat{\pi}_b \to \Delta\hat{w}_{e/b} \to 0$) into the here obtained bound.

*Note on assumptions:* The proof of Quasi-Oracle efficiency holds under the standard assumptions of sample-splitting (first and second stage are fit on separate parts of the dataset), i.i.d. data, well-behaved (convex) risk, sufficient convergence rates of nuisances, and boundedness of first moments. Of specific note is the i.i.d. assumptions, which we assume for ease of exposition, while actually only needing a less strict requirement of the empirical expectation concentrating around the exact population expectation. For a Markov chain induced by following the policy $\pi_b$ (a single trajectory), it is enough for it to be ergodic. Less formally but more intuitively, we simply need the *effective* sample size to be infinite in the asymptote.

For completeness, we write out the Hessians of $\phi$'s with respect to $\eta = (\pi_b, w_{e/b}, Q_{\pi_e})$. These terms are all included in the variables $C_1, \ldots, C_4$, since they do not include any differences between estimated and true nuisances. We thus have

$$\nabla_{\eta\eta}\bar{\phi}_1 = \begin{bmatrix} D_{111}\Delta^2\bar{\pi}_b(A'|S') & 0 & D_{113}\Delta\bar{\pi}_b(A'|S')\Delta\bar{Q}_{\pi_e}(S',A') \\ 0 & 0 & 0 \\ D_{113}\Delta\bar{\pi}_b(A'|S')\Delta\bar{Q}_{\pi_e}(S',A') & 0 & 0 \end{bmatrix} \tag{102}$$

$$D_{111} = 4\delta(A' = a)\left\{ R' + \gamma v_{\pi_e}(\tilde{S}') - Q_{\pi_e}(S',A') \right\} \frac{1}{\pi_b(A'|S')^3} \tag{103}$$

$$D_{113} = 2\frac{\delta(A' = a)}{\pi_b(A'|S')^2} \tag{104}$$

$$\nabla_{\eta\eta}\bar{\phi}_2 = \begin{bmatrix} D_{211}\Delta^2\bar{\pi}_b(A'|S') & D_{212}\Delta\bar{\pi}_b(A'|S')\Delta\bar{w}_{e/b}(S'|s,a) & D_{213}\Delta\bar{\pi}_b(A'|S')\Delta\bar{Q}_{\pi_e}(S',A') \\ D_{212}\Delta\bar{\pi}_b(A'|S')\Delta\bar{w}_{e/b}(S'|s,a) & 0 & D_{223}\Delta\bar{w}_{e/b}(S'|s,a)\Delta\bar{Q}_{\pi_e}(S',A') \\ D_{213}\Delta\bar{\pi}_b(A'|S')\Delta\bar{Q}_{\pi_e}(S',A') & D_{223}\Delta\bar{w}_{e/b}(S'|s,a)\Delta\bar{Q}_{\pi_e}(S',A') & 0 \end{bmatrix} \tag{105}$$

$$D_{211} = 4\pi_e(A'|S')w_{e/b}(S'|s,a)\left\{ R' + \gamma v_{\pi_e}(\tilde{S}') - Q_{\pi_e}(S',A') \right\} \frac{1}{\pi_b(A'|S')^3} \tag{106}$$

$$D_{212} = -2\pi_e(A'|S')\left\{ R' + \gamma v_{\pi_e}(\tilde{S}') - Q_{\pi_e}(S',A') \right\} \frac{1}{\pi_b(A'|S')^2} \tag{107}$$

$$D_{213} = 2\pi_e(A'|S')w_{e/b}(S'|s,a)\frac{1}{\pi_b(A'|S')^2} \tag{108}$$

$$D_{223} = -2\frac{\pi_e(A'|S')}{\pi_b(A'|S')}, \tag{109}$$

where all the constants elements $D$ are evaluated at $\bar{\eta}$.

## D.4 IDENTIFICATION

PROOF OF THEOREM 1

*Proof.*

$$\xi_{\pi_e}(s,a) \triangleq \mathbb{E}\left[R_0 + \sum_{t=1}^{\infty} \gamma^t R_t[\pi_e(\cdot|S_t)]\middle| S_0 = s, A_0 = a\right]. \tag{110}$$

$$= \mathbb{E}_{\pi_e}\left[\sum_{t=0}^{\infty} \gamma^t R_t \middle| S_0 = s, A_0 = a\right] \tag{111}$$

$$= Q_{\pi_e}(s,a) \tag{112}$$

$$= \mathbb{E}_{\pi_b}\left[R_0 + \sum_{t=1}^{\infty} \gamma^t \rho_{1:t} R_t \middle| S_0 = s, A_0 = a\right] \tag{113}$$

The first equality follows by definition, while the second equality is by consistency and unconfoundedness assumptions, and the final equality is by the weak positivity assumption.

*Technical remark:* For the last step, we must assume the rewards are bounded, $|R_t| \leq R_{\max}$, such that we can apply the dominated convergence theorem to take the infinite sum out of the expectation, apply importance-sampling style change of distribution element-wise to each $R_t$ expectation term and then collapse everything into the final formula. $\square$

PROOF OF THEOREM 2

*Proof.* We prove the identification is valid by showing that Eq. (6) is (i) observable and (ii) has a unique solution (unique up to equality almost everywhere).

For the question of observability, we first notice that the inner expectation is over a known distribution, i.e., the treatment assignment under $\pi_e$. The remaining randomness is then in the outer expectation over $R, \tilde{S}$, conditional on $S = s, A = a$. In the MDP, the reward and transition dynamics are the source of this randomness, meaning this randomness is invariant to the policy followed. We can thus freely write the RHS of Eq. (6) as

$$f(s,a) = \mathbb{E}_{\pi_b}\left[R + \gamma \mathbb{E}_{\tilde{A} \sim \pi_e(\cdot|\tilde{S})}[f(\tilde{S}, \tilde{A})]\middle| S = s, A = a\right], \tag{114}$$

And clearly, the RHS is observable.

The uniqueness of the solution of the Bellman equation is a well-known result in RL. A rigorous proof of which is available, for example, in (Sutton & Barto, 2018). Informally, defining the RHS as the Bellman operator $T^{\pi_e}$ on $f$, it is shown that this operator is a $\gamma$-contraction mapping in the space of bounded measure functions on $\mathcal{S} \times \mathcal{A}$. By the Banach fixed-point theorem, this implies that $T^{\pi_e}$ admits a unique fixed point. Since $f = Q_{\pi_e}$ satisfies Eq. (6), we have shown that $Q_{\pi_e}$ is the unique solution (up to equality almost everywhere). $\square$

## D.5 PROOF THAT $L^3_{\pi_e}$ TARGETS $Q_{\pi_e}$

For completeness, we prove that $L^3_{\pi_e}$ is minimized by $Q_{\pi_e}$.

*Proof.* We begin by reversing the square completion

$$L^3_{\pi_e}(\eta, g) \overset{\arg\min}{=} \hat{L}^2_{\pi_e}(\eta, g)$$

$$= \mathbb{E}_{O' \sim p_b}\left\{\sum_a \pi_e(a|S')\left(Q_{\pi_e}(S', a) - g(S', a)\right)^2 + 2\left\{R' + \gamma v_{\pi_e}(\tilde{S}') - Q_{\pi_e}(S', A')\right\}\frac{\pi_e(A'|S')}{\pi_b(A'|S')}\right.$$

$$\left. \times \left[Q_{\pi_e}(S', A') - g(S', A') + \mathbb{E}_{s,a \sim p_b(s)\pi_e(a|s)}\left[(Q_{\pi_e}(s,a) - g(s,a))w_{e/b}(S'|s,a)\right]\right]\right\}$$

The proof can be completed using Lemma 3 to remove the second term from the expectation (using the law of iterated expectations on $S', A'$). With only the first term remaining, we recognize $L^1_{\pi_e}$. Alternatively, we can arrive at $L^1_{\pi_e}$ by reversing the construction of $L^2_{\pi_e}$, namely that

$$L^2_{\pi_e} = \mathbb{E}_{O' \sim p_b} \left[ L^1_{\pi_e}(\eta, g) + \mathbb{IF}(L^1_{\pi_e}(\eta, g), O') \right] = \mathbb{E}_{O' \sim p_b} \left[ L^1_{\pi_e}(\eta, g) \right] = L^1_{\pi_e}(\eta, g), \qquad (115)$$

since efficient influence functions are mean zero by definition. Finally, showing that $Q_{\pi_e}$ minimizes $L^1_{\pi_e}$ is trivial. $\qquad\square$

# E  IMPLEMENTATION DETAILS

Anonymous code is available at `https://github.com/EmilJavurek/Orthogonal-Q-in-MDPs`. Upon acceptance, we move our code to a public GitHub repository. All experiments are implemented in the Taxi environment from the OpenAI Gym package (Brockman et al., 2016). Since the focus of our work is on second-stage estimation, we take the ground-truth oracle for the density ratio nuisances, while the first stage $Q$ is estimated for each method. We list all relevant hyperparameters in the following table. All experiments were conducted for 5 runs with different seeds.

| Component | Hyperparameter | Value |
|---|---|---|
| Taxi environment | $\gamma$ | 0.9 |
| | max_steps | 100 |
| Online $Q$ control *(to construct policies via $Q^*$)* | episodes | 5000 |
| | $\epsilon$ | 0.05 |
| | $\alpha$ | 0.1 |
| $\pi_b$ | $\epsilon$ | 0.5 |
| $\pi_e$ | $\epsilon$ | 0.1 |
| $\mathcal{D}_{\pi_b}$ | $n$ | 3000 |
| Ground-truth reference $Q_{\pi_e}$ online Expected SARSA prediction | episodes | 100000 |
| | $\alpha$ | 0.9 |
| 1st Stage | $\hat{\pi}_b$ | oracle |
| | $\hat{w}_{e/b}$ | oracle |
| | $\hat{Q}^1_{\pi_e}$ | FQE |
| DR-learner | iterations | 1000 |
| FQE | iterations | 50 |
| $Q$-regression | — | — |
| MQL | iterations | 500 |

Table 1: Hyperparameter settings for experiments in the Taxi environment.

