# OpenReview forum: "An Orthogonal Learner for Individualized Outcomes in Markov Decision Processes"
_ICLR.cc/2026/Conference — ICLR 2026 Poster_

### Official Review · Reviewer_Pa5y · 2025-10-24

**Soundness:** 2
**Presentation:** 3
**Contribution:** 3
**Rating:** 4
**Confidence:** 3

**Summary:**

The paper studies off-policy estimation of the full Q-function  from trajectories generated by a different behavior policy. The authors propose  DRQ-learner, a two-stage learner: Stage-1 fits “nuisance” components; Stage-2 minimizes an EIF-based loss to debias the final Q function. Theoretically, they claim Neyman-orthogonality, double robustness, and quasi-oracle efficiency. Empirically, on Taxi-like settings they vary data size, horizon, and overlap and report that DRQ outperforms plug-in baselines.

**Strengths:**

1. The EIF-based orthogonal loss directly targets $Q^{\pi_e}$, and cancels 1st-order nuisance error, well-motivated.
2. The paper is well-organized and clearly written.

**Weaknesses:**

1. Most results are on Taxi-like setups; no continuous-control tasks.
2. The paper varies $\epsilon$ in $\epsilon$-greedy policies but never provides a simple numerical overlap indicator, so readers cannot tell how low the overlap actually is.
3. Several results appear to use oracle or near-oracle nuisance components, so the practical robustness of Stage-2 under ordinary estimation errors is not fully demonstrated.

**Questions:**

1. Theorem 4 uses $\Delta k$ without defining $k$; seems like a leftover typo and should be fixed.

2. What exactly is the measure $p_b \pi_e$ in the theorem 4? Why is this weighting natural for your analysis?

---

> ### Author Response · Authors · 2025-11-20
> **Response to Reviewer Pa5y (1)**
>
> Thank you very much for your thoughtful review! Below, we address all your comments and suggestions. As a result, we made several improvements to our paper, which we highlight in **blue color** (see our revised **rebuttal PDF**).
>
> ## Response to Weaknesses
>
> ### W1: Empirical results all on Taxi-like setups
>
> To broaden the empirical evidence, we have **added new experiments** on the Frozen Lake environment (see **new Figure 6** and **expanded Section 6**). In these experiments, we vary (1) the dataset size, (2) the length of the effective horizon, as well as (3) the size of overlap. We confirm our method successfully achieves performance on par with the state-of-the-art: Importantly, our method remains **robust under limited overlap**, supporting the **practical validity of our theoretical guarantees**. In sum, this confirms the *generalizability *of our theoretical findings.
>
> **Action:** We performed **new experiments** to validate our results across more MDPs. (see **new Figure 6 and expanded Section 6**). **The experiments empirically confirm our theoretical results.**
>
> ###  W2: Overlap indicator
>
> Thank you for this suggestion.
>
> **Action:** We have added a second x-axis to the plots that explicitly shows the level of overlap (see our **revised Figures 4-6, right column**).
>
> ### W3: Empirical results use near-oracle nuisances.
>
> Thank you for the comment. Our work is primarily **theoretical**, and, therefore, our experiments are designed to **validate the theory**. We would like to emphasize that our choice of using near-oracle nuisance is there to ensure fair comparisons. The reason is that since the baselines do not enjoy the same robustness guarantees as our method. Hence, the baselines stand to benefit the most from using near-oracle nuisances, while our method is explicitly designed to remain stable even under estimation noise. Using oracle or near-oracle nuisances is a standard and fair practice in this line of work: it isolates the performance of the proposed Stage-2 estimator from errors in Stage-1 nuisance estimation. Thus, these experiments represent a *worst-case comparison* in terms of relative performance to other methods. Hence, this setup is conservative and allows for fair comparisons.
>
> ## Response to Questions
>
> ### Q1: Notation of $\Delta k$ in Theorem 4
> We apologise for the confusion. We wanted to simply define the $\Delta$ operator on any estimator $k$ (i.e., any of $g,\pi_{b},w_{e/b},Q_{\pi_e}$) as the difference between the learned fit with estimated and oracle nuisances.  \
>  \
> **Action:** We have revised the phrasing to better reflect this intention.
>
> ### Q2: Measure in norms in Theorem 4
>
>  \
> The measure $p_{b}\pi_{e}$ in the norms of Theorem 4 amounts to weighting the function evaluations in the $\mathcal{S} \times \mathcal{A}$ space according to the evaluation distribution corresponding to the potential distribution under $\pi_e$. In other words, we weigh initial states according to their observed distribution and weigh actions according to their propensity to be selected under the evaluation policy $(s,a) \sim p_{b}(s)\pi_{e}(a|s)$. This weighting mirrors the one in the weighted MSE $L_{\pi_{e}}^{1}$ (eq.24) and, while our loss $L_{\pi_{e}}^{3}$ is a little more complex than a weighted MSE, it may be interpreted to follow the same weighting (notice both terms have initial states still sampled according to $p_{b}(s)$ and actions according to $\pi_{e}$).
>
> **Action:** We have added a clarification to the text.

---

### Official Review · Reviewer_V1QJ · 2025-10-30

**Soundness:** 3
**Presentation:** 3
**Contribution:** 3
**Rating:** 6
**Confidence:** 3

**Summary:**

This paper addresses the challenge of estimating individualized potential outcomes in sequential decision-making, specifically the off-policy estimation of Q-functions in Markov decision processes (MDPs) using observational data. The authors introduce a novel meta-learner called the DRQ-learner, which is based on principles of causal inference and semiparametric efficiency theory. The key contribution of this work is a theoretically grounded framework that produces an estimator with three important properties: double robustness (the ability to provide valid inferences even when the behavior policy or the density ratio is mispecified), Neyman orthogonality (resilience to first-order errors in estimating nuisance functions), and quasi-oracle efficiency (asymptotic performance that matches that of an oracle with access to the true nuisance functions). In contrast to previous methods that either experience plug-in bias or struggle with the "curse of horizon," the DRQ-learner utilizes a debiased loss derived from the efficient influence function. This allows for stable and accurate long-horizon estimation, even in scenarios with low policy overlap. Additionally, the method is flexible, capable of handling both discrete and continuous state spaces, and supports various machine learning models, such as neural networks, for estimating nuisance functions.

**Strengths:**

The paper is strong due to its solid theoretical foundation and its clear synthesis of ideas from causal inference, orthogonal statistical learning, and off-policy reinforcement learning. The authors effectively reframe Q-function estimation as a causal estimand and identify the limitations of existing plug-in approaches, such as Q-regression and FQE, by examining plug-in bias. They also provide formal proofs of the key properties of their method. Empirical results from the Taxi environment support the theoretical claims, demonstrating consistent improvements over baseline methods, especially in scenarios with low overlap and long horizons. The findings confirm the advantages of Neyman orthogonality and the approach's applicability to both unrestricted and restricted model classes.

**Weaknesses:**

- While the theoretical guarantees are strong, the practical implementation relies on accurate estimation of complex nuisances (e.g., the stationary density ratio), which may be challenging in high-dimensional or continuous settings despite the orthogonality property.

- The experiments are limited to a single, relatively simple environment; broader validation across diverse MDPs (e.g., with partial observability or real-world medical data) would strengthen the empirical claims.

**Questions:**

- What are the standard assumptions in Theorem 4?

---

> ### Author Response · Authors · 2025-11-20
> **Response to Reviewer V1QJ (1)**
>
> Thank you very much for your thoughtful review! Below, we address all your comments and suggestions. As a result, we made several improvements to our paper, which we highlight in **blue color** (see our revised **rebuttal PDF**).
>
>
> ## Response to Weaknesses
>
> ### W1: Challenges of nuisance estimation in practice:
>
> Thank you for the comment. Our work is primarily **theoretical**, and our main contribution lies in establishing the **principled theory** and providing a **robust estimation framework** rather than proposing new practical estimators for each nuisance component. Hence, we would like to list some justifications for our approach:
>
> 1) *The second stage is agnostic to choice of nuisance estimation model.* Apart from the sufficient convergence guarantees, the practitioner is free to employ any method of their choosing to estimate the nuisance. This is especially important in cases such as ours, where some of the nuisances are more complex.
>
>
> 2) *The two-stage learner can, in principle, only ever improve upon the 1st stage*. Since our method requires the estimation of the target $Q_{\pi_e}$ in the first stage nuisance estimation, we are, of course, always free to stick with the nuisance estimate. Our DRQ learner is of use in settings where additional estimation complexity is worth it for the strong theoretical guarantees we provide in return. This motivation is natural for many high-stakes real-world applications such as medical applications listed in the Introduction.
>
>
> 3) *The nuisances are only as complex as the underlying setting*. Given the principled derivation of our method with the efficient influence function, all nuisance functions present in the loss come from the statistical model inherent to the problem setting.
>
> It is worth pointing out existing papers that propose estimation methods for similar stationary density ratio nuisances such as ours ( e.g., the work in [1]). Conversely, methods not using these complex nuisances may be deceptively simple: FQE does not use this nuisance but is well known to have unpredictable failure regimes (see “deadly triad” problem, Sutton & Barto (2018)).
>
> **Action:** We added a short Discussion (see **Appendix B**) where we discuss the challenges of nuisance estimation in practice. We also spell out that we mainly make theoretical contributions.
>
> ### W2: Broader empirical validation across more MDPs
>
> Thank you for the suggestion. Our main contribution is **theoretical** - we develop the first principled framework for efficient and orthogonal Q-estimation. The experiments are therefore intended primarily to **validate and illustrate our theoretical results**, rather than to provide an extensive empirical benchmark. \
>  \
> To broaden the empirical evidence, we have **added new experiments** on the Frozen Lake environment (see **new Figure 6** and **expanded Section 6**). In these experiments, we vary (1) the dataset size, (2) the length of the effective horizon as well as (3) the size of overlap. We confirm our method successfully achieves performance on par with the state-of-the-art: this confirms the *generalizability* of our theoretical findings.
>
> **Action:**
> We performed **new experiments** to validate our results across more MDPs. (see **new Figure 6** and **expanded Section 6**). **The experiments empirically confirm our theoretical results.**
>
>
> ## Response to Questions
>
> ### Q1: Standard assumptions in Theorem 4:
> Thank you for the thoughtful question. The standard assumptions for quasi-oracle efficiency can be found in the orthogonal statistical learning literature (Foster & Syrgkanis (2019)). Namely, they include: (i) for the loss of a higher-order smoothness, (ii) strong convexity, (iii) first-order optimality at the minimizer and, crucially, (iv) Neyman-orthogonality. Finally, (v) all nuisance functions must be bounded.
>  \
> **Action:** We have clarified the standard assumptions for Theorem 4 (see revised Section 5.1). We have also added a more thorough explanation and formal statements in the step-by-step derivations (see revised **Appendix D.3**).
>
> **References**
>
> [1] Nachum, O., Chow, Y., Dai, B., & Li, L. (2019, January 1). DualDICE: Behavior-Agnostic Estimation of Discounted Stationary  Distribution Corrections. *NeurIPS*.[ http://arxiv.org/pdf/1906.04733](http://arxiv.org/pdf/1906.04733)

---

### Official Review · Reviewer_MFXq · 2025-10-30

**Soundness:** 3
**Presentation:** 2
**Contribution:** 2
**Rating:** 4
**Confidence:** 4

**Summary:**

The authors address the problem of estimating individualized potential outcomes in sequential decision-making (i.e., estimating Q-functions in Markov decision processes with observational data) through a causal inference lens. They propose a novel meta-learner, termed the DRQ-learner, which enjoys beneficial theoretical properties: double robustness, Neyman orthogonality, and quasi-oracle efficiency. Empirically, the paper demonstrates that the DRQ-learner consistently outperforms baseline Q-regression and Q-learning methods in terms of the rMSE metric across a different settings.

**Strengths:**

**Novelty intersection causal inference & orthogonal learning and MDPs.** The paper creatively bridges causal‐inference, orthogonal learning with off‐policy Q‐function estimation in MDPs. By leveraging orthogonal meta‐learner ideas, it provides a fresh perspective on learning in Markov decision processes.

**Theoretical guarantees.** The authors present comprehensive theoretical analysis, including identification from multiple perspectives, derivation of efficient influence functions, and construction of an orthogonal loss. The resulting estimator is shown to possess several theoretical properties.

**Weaknesses:**

**1. Motivation for the MDP framework.**
While the proposed orthogonal learner is formulated within an MDP setting, the motivation for adopting this framework and the necessity of the Markov structure remain unclear. It would be helpful to clarify what advantages or practical challenges the MDP formulation addresses compared with alternative settings, such as DTR or contextual bandit settings.

**2. Interpretation of theorems.**
The paper includes several lengthy and dense formulas. Some theoretical results would benefit from additional intuition and clearer interpretation to help readers understand the key ideas and implications behind the derivations.

**3. Experiments.**
The experiments demonstrate that the proposed DRQ estimator outperforms strong baselines. However, several theoretical results, such as double robustness under misspecification of $\pi_b$ or $w_{e/b}$, are not empirically verified.

See also questions below.

**Questions:**

1. The authors mention the “curse of horizon” multiple times. Could you provide an explanation of this phenomenon and briefly clarify why it arises in other estimators? Additionally, do other baseline methods in the experiments also suffer from this issue?

2. In Theorem 3, could the authors provide more intuition behind $\phi_1$ and $\phi_2$? Moreover, the connections among Eq (13)–(15) and Eq (16) are not clearly explained. The proof here is not clear and requires more interpretation.

3. In Figure B3, DRQ does not perform best when $\epsilon$ is large. How to explain it?

4. There appear to be a few typos: a misplaced comma in Equation (15); inconsistent indices used in Eq (1): first $T_i$ and subsequent $nT$; and a typographical error (“si” → “is”) in Appendix C.4, line 1283. Besides, Several equations in the Appendix are too long to be fully displayed.

---

> ### Author Response · Authors · 2025-11-20
> **Response to Reviewer MFXq (1)**
>
> Thank you very much for your thoughtful review! Below, we address all your comments and suggestions. As a result, we made several improvements to our paper, which we highlight in **blue color** (see our revised **rebuttal PDF**).
>
>
> ## Response to Weaknesses
>
> ### W1: Motivation of MDP framework:
> Thank you for giving us the opportunity to motivate why we adopt the MDP setting. Indeed, the MDP setting is very common in the RL literature and applications (see e.g. Sutton & Barto, (2019) or Puterman (1994)). Importantly, the MDP formulation is the *conventional* framework for sequential decision-making, where actions affect future states and rewards. It naturally encompasses a wide range of practical applications such as personalized treatment over time, dynamic pricing, and adaptive tutoring systems. In fact, most reinforcement learning and dynamic treatment algorithms - explicitly or implicitly - rely on a Markov structure to capture the dependence of future outcomes on the current state and action. Hence, the Markov framework should not be seen as restrictive, *but rather as a standard abstraction to formalize modelling and estimation*. \
>  \
> In practice, the MDP framework is a powerful way to make estimation across long time horizons *feasible*. The **Markov property is crucial for breaking the curse of horizon** (see e.g. Kallus & Uehara (2022) for a discussion of this phenomenon in off-policy policy value estimation) . The **curse of horizon** refers to the exponential growth in computational and statistical complexity as the decision horizon increases. The reason for this is the exponential decay in overlap between trajectories from the observed and evaluation policies Without additional structure, estimating quantities that depend on the full trajectory requires handling a state space that grows exponentially with time, making both computation and inference intractable. The Markov property addresses this challenge by assuming that the current state captures all information relevant for predicting the future. This reduces the dependency structure to a single time step, allowing algorithms to operate efficiently in both theory and practice. In this sense, the Markov assumption is not restrictive-it is what makes learning and estimation feasible in long-horizon sequential problems (see also Kallus & Uehara, 2022).
>
>
> Importantly, the Markov property is not restrictive, but it is actually very general: we can always incorporate any historical information $H$ into state variable, simply by concatenating both and creating an “augmented” state variable $\tilde{S} = (S,H)$ where $H$ is any additional historical information we wish to store beyond any natural concept of state $S$. The Markov framework will natively apply to such a setup. As such, the MDP setting can also be seen as a generalization of contextual bands and some dynamic treatment regimen (DTR) setups. Specifically, contextual bandits are a one-step MDP with no state transitions (or equivalently, all state transitions are equally possible, i.e., irrelevant). DTR setups usually operate on short time horizons since without assuming Markovianity (=MDP), they invariably also suffer from the curse of horizon. We kindly refer to Appendix A for extended related work regarding DTR and the relevant overview materials. \
>  \
> **Action:** We added new motivation to justify choice of the MDP framework for our task (see **new materials in Section 3**).
>
> ### W2: Interpretation of theorems
>
> Thank you. We followed your suggestion closely and added additional intuition to our revised paper to help readers understand the key ideas and implications of the derivations (see also our detailed responses to Q3).
>
> **Action:**
>  - We provided some intuition on the $\phi_1$ and $\phi_2$ pseudooutcomes in the proposed Neyman-orthogonal loss as well as an overall intuition for the structure of the loss and its principled derivation.
>  - We added more explanations of why existing efficient influence functions (EIFs) are not applicable to our task (see Section 5.1). Specifically, existing EIFs from policy value evaluation are for scalar estimands, while our estimand is a function, which is considerably more challenging .Thereby, we spell out more clearly why our derivations are non-trivial and where the unique contributions are located.
>  - We followed Reviewer 9GGk to improve the focus of our motivation. As such, we have shortened and simplified the technical exposition until Section 4.2, so that we have more focus on our method. We hope that the new exposition will help make it easier for readers to navigate our paper.

---

> ### Author Response · Authors · 2025-11-20
> **Response to Reviewer MFXq (2)**
>
> ### W3: Empirically verified robustness to nuisance misspecification
>
> Thank you for the comment. Our work is primarily **theoretical**, and, therefore, our experiments are designed to **validate the theory**. We would like to emphasize that our choice of using near-oracle nuisance is there to ensure fair comparisons. The reason is that since the baselines do not enjoy the same robustness guarantees as our method. Hence, the baselines stand to benefit the most from using near-oracle nuisances, while our method is explicitly designed to remain stable even under estimation noise. Using oracle or near-oracle nuisances is a standard and fair practice in this line of work: it isolates the performance of the proposed Stage-2 estimator from errors in Stage-1 nuisance estimation. Thus, these experiments represent a *worst-case comparison* in terms of relative performance to other methods. Hence, this setup is conservative and allows for fair comparisons.
>
> To broaden the empirical evidence, we have **added new experiments** on the Frozen Lake environment (see **new Figure 6** and **expanded Section 6**). In these experiments, we vary (1) the dataset size, (2) the length of the effective horizon as well as (3) the size of overlap. We confirm our method successfully achieves performance on par with the state-of-the-art: this confirms the *generalizability* of our theoretical findings.
>
> **Action:** We performed **new experiments** to validate our results across more MDPs. (see **new Figure 6** and **expanded Section 6**). **The experiments empirically confirm our theoretical results.**
>
>
>
> ## Response to Questions
>
> ### Q1: Curse of horizon:
>
> Thank you for the question: The *curse of horizon* refers to the phenomenon that estimation error in off-policy RL grows exponentially with the time horizon when the estimand depends on long sequences of actions (Kallus & Uehara, (2022)) . The reason for this is the exponential decay in overlap between trajectories from the observed and evaluation policies. This is an inherent difficulty of the setting, for all estimation tasks in the off-policy setting and any corresponding approaches. The technical challenge lies in successfully and also efficiently leveraging the time-invariant Markov property of the MDP setting to break this curse. For Q-regression, the weights compound multiplicatively across time, so variance explodes and reliable estimation becomes impossible once the horizon is even moderately long. FQE leverages Markovianity and thus breaks the curse of horizon (while suffering from other issues, see “deadly triad problem” Sutton & Barto (2018)). As a general rule of thumb for intuition, estimation methods viewing data as entire trajectories will suffer from the curse of horizon, while those that can digest the dataset partitioned into one-step transitions will not.
>
> **Action:** We added a paragraph to Appendix B that explains the curse of horizon.

---

> ### Author Response · Authors · 2025-11-20
> **Response to Reviewer MFXq (3)**
>
> ### Q2: Intuition behind $\phi_1$ and  $\phi_2$
> Thank you for the question. We are happy to provide more intuition behind our derivation (we also added a summary of the intuition below to our revised paper):
>
>
> 1) *Intuition behind $\phi_1$ and $\phi_2$*: \
> Both $\phi_1$ and $\phi_2$ are the respective targets or “pseudo-outcomes” of the MSE subcomponents of the Neyman-orthogonal loss. Each contains the $Q_{\pi_e}$ term, which, if alone, would correspond simply to the standard MSE loss without debiasing. This is the non-Neyman-orthogonal starting point that we aim to debias in order to obtain robustness wrt. Nuisance estimation error. The additional debiasing terms of both $\phi_1$ and $\phi_2$ are two variations of temporal difference error (curly brackets $R' + \gamma v_{\pi_e}(\tilde{S}') - Q_{\pi_e}(S',A')$) scaled by an importance-sampling-like density ratio.
>  \
> While the density ratios here are quite complicated, the overall form of the Neyman-orthogonal loss is not. The debiasing being of the form “mean zero error scaled by density ratio” is common across many instances of DR learners in standard causal inference, including but not limited to ATE and CATE estimation.
>  \
>  In $\phi_1$, the density-ratio is the simple one-step (first-step) inverse propensity weighting. In $\phi_2$, the density-ratio is of the conditional stationary state density. Borrowing from Markov chain terminology, $2\frac{\pi_e(A' \mid S')}{\pi_b(A' \mid S')}w_{e/b}(S' \mid s,a)$ is the ratio of the likelihood of observing the state-action pair $(S’,A’)$ following from the stationary distribution of the chain induced by following policy $\pi_e$ (conditional on the chain having begun at pair $s,a$) versus following from the initial distribution following policy $\pi_b$.
>
>  2) *How we arrive at $L^3_{\pi_e}$ in Eq (13) from the EIF in Eq (16):*\
> First, we substitute the empirical risk estimator $\hat{L}^1$ of the (theoretical) population risk $L^{1}$ with a debiased estimator $\hat{L}^2 = \hat{L}^1 + \sum_{i=1}^{n}EIF(L^1, O’_i)$ (Appendix line 945). \
> Second, since this is an unwieldy loss, we complete the square to obtain the final loss $L^3$ which is a sum of two MSEs (Appendix line 950). In theory, this is a purely cosmetic step, since adding a constant (and thus completing the square) does not change the gradient of the loss and thus neither its minimizer or fitting steps. \
> The EIF necessary for debiasing is derived in Appendix lines 815-940.
>  \
>
> **Action:** We added a summary of the above intuition above the proof to our revised paper (see **revised Section 5.1**). We also spelled out clearly that the detailed step-by-step derivation is in Appendix D.1.1
>
> ### Q3: Figure B3:
>
>
> Thank you. At large $\varepsilon$ the policy $\pi_e$ whose $Q_{\pi_e}$ we wish to estimate is approaching pure randomness. While the performance of DRQ is not bad here, it is slightly outperformed by Q-regression since this is arguably the best case scenario for it. The exponential build-up of cumulative density ratios in Q-regression that usually creates issues (see other figures) is not an issue here. The numerator of the density ratio is consistently low (since $\pi_e$ is approaching uniform randomness)  leading to either normal or low overall ratios. Small ratios exponentially decay (not explode) which in the restricted model class setting is an issue that gets smoothed out. \
> Approaching the explanation from the other end, the *advantageous properties of DRQ learner are regarding robustness to nuisance estimation error and rates of convergence*. With easier nuisance functions, their estimation is easier and robustness properties are not as critical. With rates of convergence guarantees, we have a theoretical advantage in the asymptotic behavior, which does not necessarily imply best finite sample performance across all scenarios.
>  \
> **Action:** We added a short explanation summarizing the above to our main paper (see **expanded Section 6**).
>
>
> ### Q4: Typos
>
> Thank you very much for spotting the typos!. **Action:** We have fixed all the spotted typos and tightened some of the equations in the Appendix to make them fit.

---

### Official Review · Reviewer_9GGk · 2025-10-30

**Soundness:** 3
**Presentation:** 3
**Contribution:** 2
**Rating:** 4
**Confidence:** 4

**Summary:**

The authors used a two-stage approach based on the nature of double machine learning to construct a semi-parametrically efficient estimator for the Q function under the MDP setting, which is helpful for individualized and robust decision making. The authors claim novelty by extending existing work that mainly focuses on value function estimation or is not doubly robust, to the case of doubly robust Q function estimation. The corresponding theory is provided in detail.

Overall, this is a well-structured paper, but I find it lacks theoretical novelty (or at least does not emphasize it clearly) compared with existing papers, given that DML and doubly robust OPE have been extensively studied in the last 5-10 years. I think it is a good practice of applying these theories to Q function estimation, but it may not be solid enough for publication at ICLR. I would vote for a weak reject, and I believe this paper would be a good fit for theory-extension journals or conferences, although I’m open for discussion and won’t hold a very strong opinion.

**Strengths:**

1. The paper is well structured and carefully written. The flow is clear, and the theoretical results (with a not very detailed look) generally look fine.

2. For estimation construction, the authors propose a two-stage estimation that can get rid of modeling class restrictions as in Shi et al. (2021). I think this part is novel (to my knowledge), and the authors justify the performance and doubly robust property well through simulation studies.

**Weaknesses:**

1. In general, the contents before Section 4.2 are well established in the related literature. In a standard MDP setting, it is not surprising to see these results. When I first read it, I expected some new perspectives, but it turns out the new contributions mainly start from Section 4.2. For example:

- Line 128–130: The authors claim that "these works do not focus on the MDP setting and are well known to suffer from the curse of horizon." To my knowledge, some of the works listed in this paragraph (such as Shi et al., 2021) are developed under the MDP framework. Maybe the authors meant that 'some' papers in this paragraph are not based on the MDP setting, but the sentence is somewhat misleading and should be rephrased.

- Line 251–256: "This causal lens, to our knowledge, has not been made explicit in estimating Q-functions." I think this causal perspective is already well established in related work, including papers the authors cited such as Uehara (2022), and has been used in many extensions studying settings with causal assumption violations (e.g., confounders, interference, etc). Thus, I don’t think the identification results are new or particularly contributive for this paper under a standard MDP setting.

2. Based on point 1, I think the authors should highlight more clearly the theoretical challenges in establishing semi-parametric efficiency for the Q function compared with existing work (which already has such results for the value function). I don’t think there is a strong theoretical difference, but if there is, it should be emphasized to strengthen the contribution.

3. Regarding estimation, although I appreciate the contribution of proposing a two-stage estimation process that can handle nonparametric function classes for DR-Q learner, I would like to hear more about the challenges when building upon existing work and how the authors address them.

4. For the simulation part, since the authors highlight many comparisons with Shi et al. (2021), it would be great to also show comparisons with their paper on value function estimation. Because the proposed DR-Q learner is more flexible, showing this comparison would further strengthen the contribution.

**Questions:**

See Weaknesses.

---

> ### Author Response · Authors · 2025-11-20
> **Response to Reviewer 9GGk (1)**
>
> Thank you very much for your thoughtful review! Below, we address all your comments and suggestions. As a result, we made several improvements to our paper, which we highlight in **blue color** (see our revised **rebuttal PDF**).
>
> ## Response to Weaknesses
>
> ### W1: Section 4.2 covers mainly established results
>
> Thank you for your feedback! Upon reading your comment, we realized that we spend too much attention on introducing the causal perspective to readers, rather than emphasizing our own contributions. Hence, we followed your suggestion, and, as a result, we have substantially **shortened the exposition before Section 4.2**. The earlier sections now serve only to provide concise background and context, while **we now place more emphasis on** **Section 5 onward to introduce our main contributions:** our new methods and the theoretical results.
>
> **Action:** We shortened the materials before Section 4.2 to place more emphasis on our main contributions (Section 5).
>
>  - **Ad Line 128-130**: Thank you for the suggestion. We have clarified the phrasing. We now say: existing works *on individualized treatment effect estimation over time do not focus on the MDP setting* (see our changes; now in Line 128). As you correctly point out, other works we cite in the paragraph do focus on the MDP setting; however, none of these develop a meta-learning framework for the individualized treatment effect. **The latter is our novelty.**
>  - **Ad Line 251-256:** We adjusted the lines to better align with prior causal formulations summarized in Uehara et al. (2022). As a result of our changes, we place less emphasis on the identification formulas for $Q$ and instead highlight our novel rephrasing of previous algorithms from the causal perspective (Section 4.1)
>  \
> **Action:** We followed your suggestions and moved parts of Sections 3 and 4 to the new Appendix B to have more space to present our main contributions (Section 5).
>
> ### W2 and W3: Challenges in our setting
>
> Thank you for this insightful comment. Below, we explain this in greater detail: (1) we first explain why the EIFs from prior theory do **not** apply; (2) we then explain what is **new in our derivations**; and (3) we list concrete contributions that we make as a result.
>
> 1. *Why existing semi-parametric efficiency results do not directly apply:*
> Existing work such as Kallus & Uehara (2022) establishes semi-parametric efficiency for off-policy policy value estimation. Their efficient influence function (EIF) applies to the policy value, which is a **scalar**. In contrast, our setting focuses on the Q-function, which is a **function**. This difference is crucial: since the statistical estimand of interest is different, a derivation of a new efficient influence function is necessary
>  2. *What is new in our derivations:*
> In addition to deriving a new EIF, we tackle the additional challenges connected to estimating a function. Robustifying the estimation of $Q$  introduces considerable additional complexity, namely the need for a derivation of Neyman-orthogonality losses - this step was never done before in previous work on robust estimation in MDPs (see Appendix C.2). We complement this result with the quasi-oracle efficiency of the resulting estimator (Appendix C.3)
>  3. *What contributions our work makes*
> On top of the immediate applicability of our DRQ-learner, with our theoretical results, we hope to enable others to use the EIF to derive other estimators, similar to how the work of Kallus & Uehara, (2022) spurred future work into OPE policy value estimation such as that of e.g.  Shi et al. (2021) or Farajtabar et al. (2018).
>
> **Action:** We have revised the start of **Section 5.1** to clarify the novelty of our theory, especially why existing EIFs cannot be adapted and why new derivations are needed.
>
> ### W4: Comparison against Shi et al (2021)
>
> Thank you for the suggestion. We would like to use this opportunity to spell out the differences between Shi et al (2021) and our method more clearly:
>
>  1. *Different setting:* The theory of Shi et al (2021) is applicable only for point-wise estimation; i.e. learning a look-up table, while in contrast our method generalizes to much more practical scenarios such as **restricted model class** of estimator or dealing with **continuous state space**.
>  2. *Theoretical properties*: We are the first to derive a robust estimation for $Q$ in a principled manner by deriving the efficient influence function. Since we took a principled theoretical approach to deriving an estimator, we were able to get a much more generally applicable estimator as well as actually derive properties such as Neyman-orthogonality and quasi-oracle efficiency, which were not shown before.
>
> **Action:** We revised our paper to  spell out the differences more clearly and clarify that Shi et al. (2021) is either out of scope for the experiment or is generally **not** applicable to our main experiments (see Remark in our revised Section 5.1).

---

### Author Response · Authors · 2025-11-20
**Response to all Reviewers**

Thank you very much for the evaluation of our paper and your helpful feedback! We addressed all of them in the comments below. We also uploaded a revised version of our paper (see **rebuttal PDF**), where we **highlight key changes in blue color**.

Following the feedback of the reviewers, our **main improvements** are the following:

 - **New experiments:** We **added a comprehensive set of new experiments** on the *Frozen Lake* environment to provide a broader empirical validation of our theoretical claims. In the experiments, we compare the theoretical properties of our method by varying (1) the dataset size, (2) the horizon length, and (3) the overlap level. The results confirm that our method achieves performance on par with state-of-the-art baselines and remains robust under challenging conditions, thereby **empirically supporting our theoretical guarantees** (see **new Figure 6** and expanded **Section 6)**.

 - **Clarification of theoretical novelty:** We now spell out explicitly why efficient influence functions (EIFs) from existing works are **not** applicable. Importantly, prior work (e.g., Kallus & Uehara, 2022) focuses on policy value estimation, which is a *scalar estimand*, while our work focuses on the Q-function, which is a functional estimand. Hence, this requires new orthogonal statistical learning theory.

 - **Added intuition for theoretical results:** We expanded the discussion around the main theorems to provide additional intuition and clearer explanations of the proof ideas (see revised **Section 5**).

 - **Improved organization and focus:** We **shortened and streamlined** the materials before Section 4.2 to focus the exposition on our **main theoretical and methodological contributions** in Section 5.

 - **Additional explanatory material:** We introduced a new appendix, which elaborates on background topics such as the *curse of horizon* and provides clarifying examples (see **new Appendix B**).

 - **Minor fixes.** We did many many fixes. We fixed typos, improved phrasing, and clarified ambiguous statements throughout.

Given these improvements, we are confident that our paper will be a valuable contribution to the causal machine learning literature and a good fit for ICLR 2026.

---

### Meta-Review · Area_Chair_SzMt · 2026-01-09

**Summary:**

This paper develops a two-stage, orthogonal meta-learner (DRQ-learner) for off-policy Q-function estimation in MDPs, framed through causal inference and semiparametric efficiency theory. The main contribution is a new EIF-based loss for Q, yielding an estimator that is simultaneously doubly robust, Neyman-orthogonal, and quasi-oracle efficient, and that applies to both unrestricted and restricted function classes. The authors carefully connect existing Q-regression/FQE methods to plug-in learners suffering from plug-in bias, derive the efficient influence function for the functional Q estimand (rather than scalar policy value), and prove the desired orthogonality and efficiency properties. The added Frozen Lake experiments, together with the original Taxi results, reasonably support the theory by showing DRQ behaves as predicted under varying horizon, sample size, and overlap. Overall, I agree with Reviewers V1QJ, MFXq, and Pa5y that the theoretical development is sound, nontrivial, and a worthwhile contribution at the intersection of causal inference and off-policy RL.

At the same time, several presentation and scope issues should be addressed in revision. Reviewers 9GGk and MFXq correctly note that the early sections spend too much space on standard causal/MDP preliminaries and identification, which dilutes the focus on the genuinely novel part (the EIF derivation and DRQ loss in Section 5). I would encourage the authors to put more emphasis up front on what is new: the functional EIF for Q, the construction of the orthogonal loss, and the resulting properties. In addition, the empirical section remains limited to relatively simple Gym environments and near-oracle nuisance estimation; future versions should better acknowledge this and, if possible, extend to richer MDPs or more realistic settings where nuisance estimation is nontrivial. Despite these limitations, I believe the core theoretical contribution is strong enough to merit acceptance.

**Reviewer Concerns:**

Reviewer MFXq.
The rebuttal addressed several of MFXq’s concrete issues. The authors clarified why they adopt the full MDP framework (rather than a simpler DTR or contextual bandit formulation) and added text explicitly motivating the Markov structure in terms of sequential decision problems and the curse of horizon. They also expanded Section 5.1 with more intuition for the EIF-based loss and added a pointer to a detailed, step-by-step derivation in the appendix, together with fixes for the noted typos and notational inconsistencies. Finally, they added discussion and figures explaining finite-sample behavior in cases such as Figure B3. However, MFXq’s broader concerns remain only partially addressed: the paper still presents dense formulas with relatively little high-level explanation in the main text, and key theoretical properties such as double robustness under misspecification are not empirically stress-tested beyond relatively simple Gym environments.

Reviewer 9GGk.
The rebuttal responds to 9GGk’s comments by tightening the connection between the causal-inference framing and the construction of DRQ, clarifying how existing off-policy estimators fit into their plug-in taxonomy, and explaining more clearly that their main novelty lies in deriving the efficient influence function for a functional Q estimand and using it to build an orthogonal meta-learner. They also emphasize that the experiments are meant to validate theoretical predictions (e.g., behavior under low overlap) rather than to claim state-of-the-art RL performance. Nonetheless, some of 9GGk’s reservations remain outstanding: the empirical section is still limited to relatively small, tabular environments with near-oracle nuisance estimation, and the paper still spends substantial space on standard background before reaching the core EIF/DRQ contributions, which somewhat obscures the novelty.

Reviewer Pa5y.
For Pa5y, the authors clarified why a new EIF is needed in this setting (functional Q rather than scalar value), added more intuition around the orthogonal loss, and expanded the experimental section (e.g., additional Frozen Lake results) to better illustrate DRQ’s behavior under varying horizon and overlap. These changes address most of the reviewer’s technical questions. What remains, however, is the scope of the empirical validation: Pa5y’s suggestion to demonstrate DRQ in richer or more realistic MDPs is acknowledged but not realized in the current version, so the practical impact of the theory is still demonstrated only on relatively simple benchmarks.

Reviewer V1QJ.
The rebuttal reinforces the causal-inference perspective that V1QJ already found appealing and clarifies the interpretation of DRQ as a two-stage orthogonal meta-learner. The authors also respond to suggestions about emphasizing their main ideas by re-organizing some material and pointing more aggressively to the appendices for standard results. Still, V1QJ’s concerns about balance between background and new content and about the limited complexity of the experiments remain only partially resolved: the theoretical core is strong, but the paper could do more to foreground it and to demonstrate DRQ beyond toy domains.

Reviewer 9GGk
Across reviews, there is a shared request for better exposition and stronger, more diverse empirical demonstrations. The rebuttal makes noticeable progress on clarity—adding intuition, fixing notation, and pointing to detailed derivations—but it cannot fully change the fact that the current experimental suite is narrow and that some readers still find the main text heavy. These outstanding issues are largely about presentation and scope rather than correctness; they do not undermine the core theoretical contribution, but they do limit how accessible and “finished” the paper feels in its current form.

**Reviewer Scores:**

Reviewer MFXq.
MFXq was generally positive on the core theory but wanted clearer intuition and more careful presentation. The rebuttal directly strengthened those aspects without revealing any new flaws. My guess is that MFXq would either keep their score as is or bump it up slightly, reflecting increased confidence in the soundness and clarity of the approach.

Reviewer 9GGk.
9GGk recognized the value of the orthogonal framing but was cautious about how clearly the novelty was framed and how limited the experiments were. The rebuttal sharpened the positioning and clarified the contribution, while acknowledging experimental scope as a limitation. I would expect 9GGk to maintain a positive score, with a small chance of a modest increase, not a decrease.

Reviewer Pa5y.
Pa5y was already favorable, viewing the theoretical development as nontrivial and well-aligned with causal-inference ideas, and mainly asked for more intuition and examples, which the authors provided. Given that their core concerns were presentation rather than correctness, I think Pa5y would likely keep the same positive score or slightly raise it after reading the rebuttal.

Reviewer V1QJ.
V1QJ’s review was also positive, emphasizing the appeal of the orthogonal-learner perspective, with suggestions about rebalancing background vs. new content and modestly expanding experiments. The rebuttal went in that direction without changing the technical substance. I would expect V1QJ to stick with their original positive score, or potentially increas

---

### Decision · Program_Chairs · 2026-01-26

Accept (Poster)